# Sea ice-related halogen enrichment at Law Dome, coastal East Antarctica

Paul Vallelonga[1*], Niccolo Maffezzoli[1], Andrew D. Moy[2,3], Mark A.J. Curran[2,3], Tessa R. Vance[3], Ross Edwards[4], Gwyn Hughes[4], Emily Barker[4], Gunnar Spreen[5], Alfonso Saiz-Lopez[6], J. Pablo Corella[6], Carlos A. Cuevas[6] and Andrea Spolaor[7,8]

[1]Centre for Ice and Climate, Niels Bohr Institute, University of Copenhagen, Juliane Maries Vej 30, Copenhagen Ø 2100 Denmark.
[2]Australian Antarctic Division, Channel Highway, Kingston, Tasmania 7050, Australia.
[3]Antarctic Climate and Ecosystem Cooperative Research Centre, University of Tasmania, Private Bag 80, Hobart, Tasmania 7001, Australia.
[4]Physics and Astronomy, Curtin University of Technology, Kent St, Bentley WA 6102, Australia.
[5]University of Bremen, Institute of Environmental Physics, Otto-Hahn-Allee 1, 28359 Bremen, Germany.
[6]Department of Atmospheric Chemistry and Climate, Institute of Physical Chemistry Rocasolano, CSIC, Serrano 119, 28006 Madrid, Spain
[7]Ca'Foscari University of Venice, Department of Environmental Sciences, Informatics and Statistics, Via Torino 155, 30170 Venice Mestre, Italy.
[8]Institute for the Dynamics of Environmental Processes, IDPA-CNR, Via Torino 155, 30170 Venice Mestre, Italy.

*Correspondence to*: Paul Vallelonga (vallelonga@nbi.ku.dk)

**Abstract.** The Law Dome site is ideal for the evaluation of sea ice proxies due to its location near to the Antarctic coast, regular and high accumulation throughout the year, an absence of surface melting or remobilization, and minimal multiyear sea ice. We present records of bromine and iodine concentrations and their enrichment beyond seawater compositions and compare these to satellite observations of first year sea ice area in the 90-130° E sector of the Wilkes coast. Our findings support the results of previous studies of sea ice variability from Law Dome, indicating that Wilkes coast sea ice area is currently at its lowest level since the start of the 20th century. From the Law Dome DSS1213 firn core, 26 years of monthly deposition data indicate that the period of peak bromine enrichment is during Austral spring-summer, from November to February. Results from a traverse along the lee (Western) side of Law Dome show low levels of sodium and bromine deposition, with the greatest fluxes in the vicinity of the Law Dome summit. Finally, multidecadal variability in iodine enrichment appears well correlated to bromine enrichment, suggesting a common source of variability that may be related to the Interdecadal Pacific Oscillation (IPO).

**Keywords.** Antarctica, Halogens, Bromine, Iodine, Law Dome, sea ice, sea ice reconstruction, polar halogen chemistry

# 1 Introduction

Atmospheric halogen chemistry is highly complex and has been a topic of intensive study over the past three decades (Saiz-Lopez and von Glasow, 2012; Simpson et al., 2015). A major branch of this research involves the occurrence of enhanced concentrations of boundary-layer bromine in coastal Antarctica, which has been linked to the depletion of tropospheric ozone and mercury (Simpson et al., 2015). Tropospheric ozone is critical to solar forcing, ultravioletUV absorption and aerosol formation in the polar regions whereas the concentration of mercury in polar snow is a matter of great concern for Arctic ecosystems in the future (Brooks et al., 2006; Hylander and Goodsite, 2006). Recent studies indicate that boundary layer bromine is driven by photochemical recycling above the salt-rich snow and ice surfaces, with such recycling predominantly occurring in the Austral spring, when the concentration of surface salts and surface area of first-year sea ice are at their greatest (Pratt et al., 2013; Zhao et al., 2016). Photochemical recycling of bromine primarily involves heterogeneous reactions of halide salts (such as HOBr and $BrONO_2$) in sea ice and snowpack leading to the emission of $Br_2$ molecules. $Br_2$ is then photodissociated into two $Br^-$ radicals that are available for further heterogeneous chemical recycling. Bromine explosion events primarily occur in early spring and summer, although winter sources of organohalide emissions have also been observed in coastal polar regions although the relative influence of such sources is still a topic of investigation (Impey et al., 1997; Nerentorp Mastromonaco et al., 2016; Simpson et al., 2007).

The processes and physical conditions underlying photochemical bromine recycling events, commonly called 'bromine explosions', are subject to investigation through a combination of satellite and insitu observations as well as chemical transport modelling. Bromine and sodium in coastal Antarctic air are predominantly sourced from sea salts, but only bromine is enriched by bromine explosion events. Processes and precursors of halogen recycling in polar regions include: the role of acids in snowpack in enhancing heterogeneous bromine production; the role of organic molecules as sources of radicals – particularly in the Arctic, where formaldehyde and other organic pollutants may be present in significant concentrations; and the partitioning of bromine between gas and particulate phases in snowpack and boundary layer air. The transport of bromine is being understood through a combination of ground and satellite-based measurements combined with simple models that incorporate explicit snowpack chemistry (Spolaor et al., 2016b; Zhao et al., 2016). These models demonstrate the importance of such features as wind velocity and vertical mixing (boundary-layer turbulence) as well as the surface parameters mentioned above (Toyota et al., 2014). Insitu observations of halogens in both gaseous and aerosol forms (Legrand et al., 2016) are essential to understanding halogen deposition and recycling processes in polar regions and improve current atmospheric models.

Reactive iodine has a global influence on ozone depletion and the oxidizing capacity of the polar atmosphere by influencing the repartitioning of $HO_x$ and $NO_x$ (Saiz-Lopez et al., 2012). In particular, large amounts of oxidized iodine detected by a ground-based spectrometer were observed in coastal Antarctica (Saiz-Lopez et al., 2007) highlighting these coastal areas as

iodine hotspots. Enhanced sea ice bioproductivity during springtime and winter organic emissions have been suggested as the main sources of iodine in the Antarctic Peninsula (Atkinson et al., 2012; Granfors et al., 2014; Saiz-Lopez et al., 2015). Recent iodine instrumental measurements in the Antarctic coastal sea ice zone showed that $I_2$ dominates the iodine atom fluxes to the atmosphere (Atkinson et al., 2012). The same study revealed that iodocarbon concentrations above sea ice
brines were over ten times greater than those of the sea water below.

The first measurements of bromine and iodine species (bromide $Br^-$ and bromate $BrO_3^-$, Iodide $I^-$ and iodate $IO_3^-$) in Antarctic ice were reported by Spolaor et al., (2012) indicating relatively stable concentrations of 100-200 pg $Br^-$ $g^{-1}$ and 5-32 pg $I^-$ $g^{-1}$ in Talos Dome ice core from the early Holocene. Bromate and iodate were not present above detection limits (38 pg $BrO_3^-$ $g^{-1}$,
7 pg $IO_3^-$ $g^{-1}$) in these samples. Subsequently, total bromine concentrations in the Talos Dome ice core were reported for the past two glacial cycles, observing a depletion of bromine relative to the Br/Na ratio found in seawater during the last two glacial maxima (Spolaor et al., 2013b). The temporal variability of bromine depletion corresponded well with a reconstruction of sea ice duration in the Victoria Land sector (Crosta et al., 2004), opening a basis for the further investigation of links between sea ice extent and bromine in polar ice cores. The seasonal nature of bromine enrichment in
Antarctic ice was demonstrated by Spolaor et al. (2014), who reported spring/summer peaks of bromine enrichment and winter peaks of iodine concentration in Law Dome ice dating to 1910-1914 CE.

Reconstruction of Antarctic sea ice from ice core-based proxies is primarily based on fluxes of sodium and/or methanesulphonic acid (MSA, $CH_3SO_3^-$) (e.g., Abram et al., 2013; Curran et al., 2003). In the EPICA Dome C ice core,
sodium concentrations were investigated by Rothlisberger et al. (2010) and compared to sea ice reconstructions from Southern Ocean marine sediment cores. For the last two glacial cycles, good correspondence was found between sea salt-derived sodium and the marine record. For reconstruction of sea ice area on decadal or even centennial scales, the influence of meteorological and depositional noise has been shown to be an important factor to consider especially for drill sites located far inland such as Dome C (Levine et al., 2014).

In contrast, the presence of MSA in ice cores has been successfully linked to observed sea ice variability in some Antarctic ice cores (e.g., Abram et al., 2007; Abram et al., 2013). In some cases, the presence of MSA is either positively or negatively correlated with sea ice, depending on such factors as local wind direction, polynya and sea ice seasonality (Abram et al., 2013). The longest reconstruction of sea ice extent using MSA has been reported for Law Dome, East Antarctica (Curran et
al., 2003) covering the past 200 years and indicating a steady decrease in sea ice extent to the year 2000 C.E. Recent satellite observations report a slow increase in Antarctic sea ice over the past three decades (Parkinson and Cavalieri, 2012), and other MSA-based sea ice reconstructions support such observations (Thomas and Abram, 2016).

In this work, we report halogen deposition from the Dome Summit South (DSS), Law Dome ice core record covering the 20[th] century, with an emphasis on high-resolution measurements corresponding to the period since satellite observations of sea ice began. We examine correlations between sea ice area and bromine as well as halogen enrichment in the DSS Law Dome ice cores. The high rate of snow accumulation and regular year-round snowfall at Law Dome makes the site ideal for such a study, as well as the detailed MSA-based sea ice reconstruction already reported for the site. Although MSA originates from a different emission source than bromine, both are related to sea ice area and hence should be expected to produce an overall similar pattern of sea ice variability on a multi-year or decadal scale.

## 2. Methods

### 2.1 Sample collection

#### 2.1.1 Dome Summit South (DSS), Law Dome

The DSS ice cores are from Law Dome, East Antarctica. Law Dome is a small ice cap located approximately 120 km inland from the Wilkes Land coast (Fig. 1). The summit of Law Dome experiences relatively high and constant year-round precipitation, providing clear seasonal variability in stable isotopes of water and ionic species (Curran et al., 1998; McMorrow et al., 2004). The summit of Law Dome features a snow accumulation rate of approximately 60 cm ice equivalent per year, with an average annual temperature of -20° C. The precipitation is predominantly due to westerly cyclonic systems which produce a strong accumulation gradient from East (high accumulation) to west (low accumulation). The Law Dome summit is at an altitude of 1310 m asl, above approximately 1200 m of ice dating back to the last glacial (Morgan et al., 1997). The ice core samples were collected from the DSS site that is within a few hundred metres of the Law Dome summit, with the specific co-ordinates and sampling details described in the following sections. In all cases, the samples described here have been dated by identification and counting of annual cycles in stable isotopes of water (Morgan and van Ommen, 1997; Roberts et al., 2015) and ionic species (Curran et al., 1998) as well as synchronisation of well-dated volcanic eruption tie points (Plummer et al., 2012). Impurities in Law Dome snow are dominated by sea salt inputs (98% by mass) with a minor input from mineral dust (2% by mass) (Vallelonga et al., 2004).

#### 2.1.2 DSS0506

A 258 m deep ice core was drilled using a 20 cm diameter thermal drill near to the Law Dome summit (66°46'19.68" S, 112°48'25.60" E) in November 2005 (Burn-Nunes et al., 2011). The core, designated DSS0506, was subsampled in October

2006 in a cold laboratory in a storage freezer in Hobart. For each ~1 metre long section of ice, a 35 mm x 35 mm stick was collected for chemical analysis, and subsequently decontaminated by chiselling with a microtome blade in an HEPA-filtered laminar flow bench in a freezer. All equipment used for the decontamination was repeatedly cleaned and stored in deionised water (Millipore MQ system, 18.2 MW/cm). The decontaminated samples were collected in polystyrene "Coulter counter" accuvettes and then melted for analysis by ion chromatograph. The remaining melted sample, usually 3 to 8 mL, was then refrozen and stored. In 2014 the remaining samples were sent to the University Ca' Foscari of Venice for bromine determination. The depth range of DSS0506 samples reported here is from 19.7 m (1991 C.E.) to 72.5 m (1929 C.E.). The sampling resolution increased with depth, from 5 cm near the surface to 3 cm at depth, ensuring higher than monthly resolution at approximately 25 and 15 samples per year, respectively.

### 2.1.3 DSS1213

In austral summer 2012/2013, a 30 m firn core was drilled at DSS, Law Dome (66°46'21" S, 112°48'40.59" E) using the 4-inch Kovacs Mark VI coring system (Roberts et al., 2015). The firn core was transported to Hobart, Australia, and subsampled in the freezer at the Australian Antarctic Division (AAD)/Antarctic Climate and Ecosystem CRC (ACE CRC). For each 1-metre length of firn, a 35 mm x 35 mm stick was cut and sent to the Trace Research Advanced Clean Environment (TRACE) laboratory at Curtin University of Technology, Perth, Western Australia, for bromine determination. A parallel 35 mm x 35 mm stick was decontaminated and analysed for trace ions at the AAD/ACE CRC in Hobart, with a parallel section analysed for stable isotopes of water. Data covering the full depth range of the firn core are reported here, covering the period from summer 2012/2013 to late 1987 and overlapping with the DSS0506 record.

### 2.1.4 Law Dome traverse and DSS1516 snowpit

Another sampling expedition was conducted at Law Dome summit in February 2016, with snow surface samples collected during the traverse to the DSS site and from a 1-m snowpit. The DSS1516 snowpit (66°46'23.51" S, 112°48'40.59" E) was collected upwind from the traverse camp (designation: Waypoint A, 66°46'22.12" S, 112°48'28.19" E) and sampling was conducted immediately after the pit was prepared. Samples were collected every 3 cm by plunging pre-cleaned 50 mL polyethylene tubes into the pit wall. Stable isotopes of water ($\delta^{18}O$ and $\delta D$) measured in the parallel snow pit wall confirm that the 1 m sequence covers the period from winter (July) 2015 to February 2016.

In addition to the DSS1516 snowpit, surface snow samples were collected during a 1-day traverse from Casey station to the DSS Campsite A. Details of the sampling sites are included in Table S1 (supplementary material). Eleven samples were

collected during the 114 km traverse, extending from an altitude of 500m asl to 1320 asl at the DSS1516 site. The samples were collected every 10 to 15 km by moving approximately 15 m upwind of the Hagglunds traverse vehicle and plunging a polyethylene tube into the snow surface. The tubes were immediately sealed after sampling and were kept frozen until analysis. All samples collected during the Casey-Law Dome traverse and from the DSS1516 snowpit were sent to the TRACE laboratory at Curtin University of Technology for bromine analysis.

## 2.2 Halogen measurements

### 2.2.1 Australia

The DSS1213 1-metre sticks were cut and prepared in the TRACE laboratory located at Curtin University of Technology, Perth, Australia (Ellis et al., 2015). The 35 mm x 35 mm cross-section sticks were melted on a silicon carbide melter head at a melt rate of 5 cm/minute following the setup of McConnell et al (2002), with a depth resolution of less than 1 mm. The central melt line was directed to a Thermo Element XR ICP-SFMS fitted with cyclonic peltier-cooled spray chamber (2° C) and 400 uL/minute PFA ST nebulizer (both from ESI, Omaha, USA). The sample line was acidified with 2% ultrapure nitric acid and internal standard (4 ppb $^{115}$In) immediately before introduction to the plasma. Only bromine and sodium were determined, at medium resolution (~4000), in the DSS1213 firn core.

Law Dome traverse and DSS1516 snowpit samples were analysed discretely using a Seafast-II autosampler with syringe pump connected to the abovementioned analytical system. For these samples, iodine ($^{127}$I) was determined at low mass resolution (~300) and all other elements at medium resolution ($^{23}$Na, $^{35}$Cl, $^{79}$Br). A 4 ppb $^{115}$In internal standard was used at both mass resolutions. Analytical blanks and quality control standards were determined after every 10 samples analysed.

Irrespective of the sample delivery method, the analytical performance of the Element XR ICP-SFMS was consistent during the measurement campaigns. Calibration standards were prepared by sequential dilution from primary stock solutions (Bromine, 10 μg/mL in $H_2O$; Sodium, 1000 μg/mL in 1% $HNO_3$) from High-Purity Standard (Charleston, USA). Sodium and bromine were calibrated using 7 concentration standards of increasing concentration up to 100 ppb (sodium) and 4 ppb (bromine). All linear calibration regressions showed $R^2$ >0.99 (n=8, p<0.001). Detection limits for bromine and sodium were 0.18 and 1.1 ppb respectively (n=80) Repeatability of measurements was systematically tested in each analytical run by measuring replicates of quality controlled standards. The variability was calculated as the standard deviation of the signals and found to be 9% for bromine and 4% for sodium.

**2.2.2 Italy**

The DSS0506 core samples were measured at the Environmental Analytical Chemistry laboratory of the University Ca'Foscari of Venice. The samples were stored in plastic vials and were kept frozen until analysis. For the DSS0506 samples, some of the polystyrene accuvettes were broken in transport and were found to be contaminated upon comparison of sodium concentrations measured before and after transport. Consequently, data corresponding to the years 1945-6, 1950-1, 1954-5 and 1963-4 are incomplete.

Similar to the Curtin University analysis, samples were measured on a Thermo Element2 ICP-SFMS instrument using a cyclonic peltier-cooled spray chamber (ESI, Omaha, USA). System cleaning and operational parameters have been described previously (Spolaor et al., 2013b) and will be summarized here. Before beginning each analytical session, the sample introduction system was cleaned with alternating washes of 5% $NH_4OH$, then 2% $HNO_3$ acid, separated by 30s of MQ water. During the analytical sessions, the sample line was thoroughly cleaned using 2% nitric acid (120 s) and UPW (120 s) between each analysis. Elements were determined in low- ($^{127}I$) and medium-resolution ($^{23}Na$, $^{79}Br$) with plasma stability evaluated by the continuous monitoring of $^{129}Xe$. External standards, ranging from 0.01 to 4 ppb, were prepared by diluting a 1000 ppm stock IC solution (TraceCERT® purity grade, Sigma-Aldrich, MO, USA). Excellent precision was found with calibration correlations of $R^2 > 0.99$ (N=6, p=0.05). Calculated detection limits were 0.05 ppb Br, 0.005 ppb I and 0.8 ppb Na.

The reproducibility of measurements between the two laboratories was tested by analyzing 140 Greenland snow pit samples in both laboratories. Compatibility of the measurements (Supplementary Figures S1 and S2) showed a regression line with $R^2 > 0.9$ (n=140, p<0.05) for both analytes. Distributions of residuals show an average measurement offset of -0.64 ± 0.19 ppb (sodium, RSD=2.0 ± 0.2 ppb) and -0.03 ± 0.01 ppb (bromine, RSD=0.11 ± 0.01 ppb).

**2.3 Ion chromatography measurements**

Major and minor ions were determined at the AAD/ACE CRC laboratory in Hobart Australia following the established suppressed ion chromatographic methods (Curran and Palmer, 2001). Samples were stored and prepared in an HEPA-filtered cleanroom and all laboratory apparatus was cleaned using filtered deionized water (Siemens Ultrapure water system, 18.2 M$\Omega$/cm). Detection limits for MSA and Na were 0.095 ppb and 0.23 ppb, with reported precisions of 3.8 ppb and 0.46 ppb and analytical ranges of 0.6-40 ppb and 0.7-500 ppb, respectively. The MSA data covering the period 1920-1995 are those reported by Curran et al. (2003). Sodium data from the DSS main ice core record have been previously discussed by Palmer et al. (2001).

**2.4 Sea ice area**

Satellite-based observations of sea ice area and concentration have been used to evaluate the suitability of bromine and
iodine as proxies for sea ice area reconstructions. We have calculated sea ice area as the product of sea ice concentration and
grid cell size for each grid cell in the two sectors considered. Note that sea ice *area* is different to, and slightly less than, sea
ice *extent*, as sea ice *extent* is commonly defined as the sum of grid cells containing at least 15% sea ice coverage. The sea
ice area has been evaluated in two ocean sectors adjacent to Law Dome and are used for comparison: one to the west of Law
Dome (90 to 110° E) and one to the east of Law Dome (110 to 130° E); as shown in Fig. 2. A strong East-to-West
accumulation gradient has been observed across Law Dome. Hence it is expected that sea salts carried to Law Dome will
predominantly originate from low pressure systems developing from the 90-110° E sector.

A discontinuous record of monthly sea ice area is calculated from a series of satellite-based sensors, covering the period
from 1974 to 2015 with missing years for 1975, 1977 and 1978. Sea ice area for the period 1973 to 1977 was determined
from daily brightness temperatures monitored by the ESMR instrument mounted on the NIMBUS-5 satellite (Parkinson et
al., 2004). For the period January 1979 to May 2015, sea ice observations were obtained from the SMMR and SSM/I and
SSMIS instruments mounted on various satellite platforms (Cavalieri et al., 1996, updated yearly). These data are freely
available from the US National Snow and Ice Data Centre (NSIDC) website (Cavalieri et al., 1999; Fetterer et al.).

First year sea ice (FYSI) area has been calculated as the difference between the late-summer (February/March) minimum and
late-winter (September/October) maximum of sea ice area occurring each year. FYSI data series' calculated for the 90-110°
E and 110-130° E sectors are shown in Fig. 3. Also shown in Fig. 3 are two other publicly available datasets reporting sea ice
extent (not sea ice area), largely calculated from the same satellite observations. The first dataset, here referred to as 'Jacka
SIE', reports monthly observations of the northernmost sea ice edge (SIE) for 10° sectors of longitude around Antarctica
over the period 1973-1998. The data are available online (https://data.aad.gov.au/metadata/records/climate_sea_ice) and
were employed by Curran et al. (2003) to validate their MSA-based Law Dome sea ice reconstruction. The second dataset is
produced by National Snow and Ice Data Centre (NSIDC) and offers near-daily sea ice extent over the period 1978-2013 for
Antarctica. The data are divided into 5 Antarctic sectors, of which the "Pacific Ocean" sector encompasses 90-160° E and
includes Law Dome. The Jacka SIE and NSIDC datasets are shown to demonstrate their general correspondence with the
FYSI data reported here for regression analysis with Law Dome bromine data. In Fig. 3, we also plot the sum of the 90-110°
E and 110-130° E FYSI datasets to allow better comparison to Jacka SIE and NSIDC.

## 3. Results and discussion

### 3.1 Halogen and sodium records

Mean bromine, iodine and sodium glaciochemical concentrations (1927-2012 CE for Br and Na, 1927-1989 CE for I) at DSS were comparable to previously reported values for Law Dome (Curran et al., 1998; Spolaor et al., 2014). Considering annual averages over the period 1927-2012, we find average concentrations of 88±45 (1σ) ng g$^{-1}$ for Na and 2.9±2.7 (1σ) ng g$^{-1}$ for Br. For iodine, the average for the period 1927-1989 is 0.061±0.023 ng g$^{-1}$ (1σ, n=62). As may be expected from the "spiky" nature of the data, both the medians (81.1 ng Na g$^{-1}$, 2.1 ng Br g$^{-1}$, 0.056 ng I g$^{-1}$) and geometric means of the data (81.0 ng

Na g$^{-1}$, 2.3 ng Br g$^{-1}$, 0.057 ng I g$^{-1}$) are 5 to 20% lower than the arithmetic means. Annually-averaged non-sea salt Br (nssBr) concentrations average 2.4 ng g$^{-1}$ (geomean 1.6 ng g$^{-1}$) are comparable to values reported previously for Law Dome (Spolaor et al., 2014). Sodium concentrations are also in good agreement with previously reported values of 3.47±3.03 (1σ) μeq L$^{-1}$ (80±70 ng g$^{-1}$) (Curran et al., 1998).

The bromine, iodine and sodium time series records determined for DSS firn and ice cores are shown in Fig. 4. No significant trend is present in any of the timeseries. The sodium data are representative of sea salt inputs from the Southern Ocean to the site whereas bromine is known to be subject to enhancement in photochemical 'bromine explosion' events. We quantify the strength of such photochemical enrichment by calculating the bromine enrichment (Br$_{enr}$) beyond the bromine/sodium abundance ratio found ubiquitously in seawater [Br/Na=0.006; Turekian (1968)]. Following common

practice (e.g. Spolaor et al., 2013a; Spolaor et al., 2016a) we calculate Br$_{enr}$ = Br$_{conc}$/(Na$_{conc}$*0.006), where Br$_{conc}$ and Na$_{conc}$ respectively describe the concentrations of Br and Na measured in a sample. For iodine enrichment (I$_{enr}$) we apply the same enrichment calculation but using the iodine/sodium ratio in seawater of 5.93 x 10$^{-6}$ (Turekian, 1968). Iodine and bromine enrichment data and their relation to sea ice variability will be discussed in the following sections.

There appears to be less interannual variability in the DSS1213 Br record, with respect to the DSS0506 record. The average values of Br$_{enr}$ and Br$_{conc}$ are similar for the two cores but the variance of annual averages is much greater for the DSS0506 samples (Table 1). For sodium, there is no substantial difference in average concentration but the opposite trend is found for variance, with greater variance in the DSS1213 data. We attribute this difference in variance to the different sampling and analytical techniques applied to each core – DSS0506 was sampled by discrete chiselling and DSS1213 sampled by

continuous melting. The ice core melter system had a flow path ~5 times longer than the discrete sampling system, and there was likely greater memory-effect of bromine coating onto the inside of the ICP-SFMS sample introduction system (peltier-cooled cyclonic spray chamber) used for the continuous melting analysis. Note that the memory effect is much more

significant for bromine than for sodium, which is efficiently transported through the ICP-SFMS spray chamber in a $HNO_3$ acid solution. Sodium does not have a significant memory effect and the higher variance found by the ice core melter is expected due to the higher depth-resolution of the melter system compared to discrete sampling. Despite the apparent aliasing of the DSS1213 Br record, the two records show good agreement in the overlap period from calendar 1988 to 1989 (Fig. 4). A double peak in Br was found with the initial peak in the DSS0506 core leading the DSS1213 record by approximately 2 weeks (from comparison of the time scales), which is well within the sub-annual dating error. Furthermore the seasonality of bromine and sodium measured in the DSS1213 samples (discussed in section 3.5) are in agreement with previous studies conducted on Law Dome samples (Curran et al., 1998; Spolaor et al., 2014).

## 3.2 Bromine and first year sea ice

Considering the known relationships between emissions of bromine and seasonal sea ice area, we investigated the reliability of bromine, and bromine enrichment, as a proxy for regional seasonal sea ice area adjacent to Law Dome. With the intention of reducing data autocorrelation, we transform the $Br_{enr}$ data to a gaussian-like distribution using the natural logarithm of $Br_{enr}$ for correlation to FYSI data. An additional reason for applying a logarithmic transformation is due to the exponential nature of the "bromine explosion" process occurring above the bromine-rich FYSI surface: in each stage of the explosion, one reactive bromine species (HOBr) is consumed in the process of liberating $Br_2$ from the sea ice surface. As $Br_2$ is the reactive precursor to two subsequent bromine explosion multiphase reactions, the process develops exponentially. Histograms of Br concentration, $Br_{enr}$ and $ln(Br_{enr})$ are shown in Supplementary Figure S3, demonstrating that the distribution of $ln(Br_{enr})$ values in Law Dome is well represented by a gaussian curve. Correlation tests have been performed using different subsets of the $ln(Br_{enr})$ data; such as summer-summer (calendar year), winter-winter (July-June) and spring-only (August-October) intervals. The FYSI data used for correlation has been described in Sect. 2.4. Note that the optimal correlation is found when $ln(Br_{enr})$ is compared to the FYSI value from the previous year. This is due to the timing of formation of sea ice in the Antarctic (from March to September) occurring, by necessity, earlier than the springtime bromine explosion that occurs in the following calendar year.

A summary of the correlation analysis between Law Dome bromine enrichment and FYSI is shown in Table 2. We observe firstly that $ln(Br_{enr})$ is better correlated with the 90-110°E sector than for the 110-130°E sector. Such a finding is consistent with the westerly circulation around Antarctica, despite the individual cyclonic systems producing an east-to-west deposition gradient across Law Dome. The 80-140° E sector was used for correlation between sea ice extent and MSA concentrations at Law Dome (Curran et al., 2003), with the best correlation found for sea ice in the 110ºE sector. The strongest correlation ($r^2$ 0.357, p<0.001) is between $ln(Br_{enr})$ and 90-110° E sector FYSI for the summer-summer calendar year period. A weaker and less-significant correlation is found between $ln(Br_{enr})$ and 110-130° E FYSI  for the winter-winter interval ($r^2$ 0.20, p<0.01).

Such a finding is counter-intuitive because the bromine explosion occurs primarily in spring/summer and should be most completely captured in the winter-winter interval. As has been demonstrated thoroughly by Levine et al. (2014), meteorological transport variability and other "noise effects" can have a strong influence on the regularity of seasonal sea salt deposition. Hence it may be that the summer-summer interval produces a marginally stronger correlation with 90-110° E FYSI due to the smoothing effect caused by the averaging of consecutive spring/summer periods.

**3.3 Bromine enrichment as a sea ice proxy**

On the basis of the significant correlation between bromine enrichment at Law Dome and 90-110° E FYSI, we consider the implications for reconstructing past sea ice area at Law Dome. Figure 5 shows Law Dome $\ln(Br_{enr})$ and MSA (Curran et al., 2003) since 1927 C.E. as well as the 90-110° E FYSI data plotted as an anomaly from the average of the data, to better display FYSI trends. We firstly note that a selection of running-mean smoothing filters has been applied to such data previously, such as 3- and 20- year means (Curran et al., 2003) and 11-year means (Abram et al., 2010). Here we follow the latter and show 11-year means as well as the individual annual data points.

When comparing two proxies purporting to represent the same phenomenon it is vital to consider the different physical processes involved in the proxy generation, transport and deposition. We note that MSA is produced biologically and, for the Law Dome sector, has been quantitatively linked to the opening of sea ice-covered seawater during the summer and autumn seasons. Bromine enrichment occurs primarily in spring/summer and is dependent upon the presence of FYSI (Saiz-Lopez and von Glasow, 2012). Due to the 11-year smoothing applied to the data, influences of seasonal patterns, factors influencing biological growth, relations to sea ice and transport efficacy should be minimised for the comparison of bromine enrichment and MSA trends at Law Dome.

Some differences are seen among the sea ice trends indicated by MSA and bromine at Law Dome over the 20[th] century (Fig. 5). Both proxies display substantial multidecadal variability, so any long-term trend is here treated with caution. In the cases of both bromine and MSA, simple linear regression indicates small declining trends for both species. Bromine enrichment values are greater during the period 1940-1950 and 1975-1985 while the highest MSA concentrations are observed during the period 1945-1955. Bromine and MSA both point toward greater sea ice area during the period from 1945 to 1950 but diverge between 1955 and 1970. The cause for this divergence is not yet known, but before speculating on a possible cause, these trends should be confirmed by measurements of other snow and ice samples from Law Dome as well as other sectors of the East Antarctic coast. The possible influence of multidecadal-scale climate variability, such as the Interdecadal Pacific Oscillation (IPO), on the bromine record is discussed in detail in section 3.4, but will briefly be considered here. IPO forcing of Antarctic sea ice area has been demonstrated at decadal timescales (Meehl et al., 2016), with the negative IPO phase

triggering SLP and near surface wind changes that can influence sea ice expansion, storm tracks and potentially nutrient supply to DMS-producing algal communities. Smaller magnitudes of change are observed in MSA over the 20$^{th}$ century, compared to Br$_{enr}$. Despite the short time period available, satellite-based observations of FYSI display positive anomalies before 1985 and negative anomalies in the last decades (Fig. 5) which are consistent with recent trends of both MSA and

Br$_{enr}$.

### 3.4 Iodine enrichment

Iodine enrichment in sea ice could be explained by complex heterogeneous reactions that take place above seasonal sea ice

releasing gas-phase iodine and particulate species. Peak I concentrations have been found in winter snow strata at Law Dome (Spolaor et al., 2014) and Neumayer station (Frieß et al., 2010), suggesting surface re-emission and/or remobilization during the Austral summer.. There is no significant correlation between Br$_{enr}$ and I$_{enr}$ signals on an annual basis (r$^2$=0.05, p<0.1). This lack of significant correlation might be related to the different iodine emissions and recycling mechanisms over sea ice: i) emission of iodine from sea-ice enhanced bioproductivity and subsequent upwards migration through brine channels (Saiz-

Lopez et al., 2015); ii) photochemical reactions over iodate frozen salts (Spolaor et al., 2012) and ii) atmospheric release of gas-phase iodine from triiodide production via iodide oxidation in frozen solution (Kim et al., 2016). Another indicator of the complex ocean-sea ice-atmosphere interrelation arises from the statistical comparison with FYSI records (Table 2). Correlation significance levels for I$_{enr}$ are consistently below the 95% level and correlation coefficients are generally low. The strongest correlation of I$_{enr}$ is between the summer-summer (calendar year) averaged signal and the 110-130° E FYSI

sector, but this is still below the 95% significance level.

Correlation between 11-year smoothed I$_{enr}$ and Br$_{enr}$ records (Fig. 6) is significant (r$^2$=0.269, p<0.001) suggesting a long-term (decadal scale) common driver controlling halogens deposition in coastal Antarctica. The iodine enrichment time series shows a similar pattern to that of bromine enrichment, with higher values particularly during the 1940s and back from negative to positive in the 1970s (Fig. 6). These periods coincide with rapid changes in the Interdecadal Pacific Oscillation

(IPO) phase from positive to negative in the 1940s and back from negative to positive in the 1970s. The IPO is a low frequency climate mode related to the El Niño-Southern Oscillation which operates on multidecadal timescales. It affects climate variability at the multidecadal scale across and beyond the Pacific Basin (Power et al., 1999; Vance et al., 2015). Impurities deposited at Law Dome have been demonstrated to faithfully reflect IPO variability (Vance et al., 2015) and

reanalysis data indicates a strong IPO signal in the Indian Ocean (Vance et al., 2016). Furthermore, recent work has demonstrated an IPO forcing of Antarctic sea ice area at decadal timescales, with the late 1990's shift to a negative IPO phase triggering SLP and near surface wind changes that are conducive and consistent with an expansion in sea ice in all seasons across multiple regions of the Antarctic seasonal ice zone (Meehl et al., 2016). Thus the overall correlation between

iodine and bromine enrichment may be linked to decadal-scale states of the atmosphere-ocean-sea ice system in the Indian sector of the Southern Ocean. It must be noted that, in addition to larger-scale influences of atmospheric transport and ocean-related sea ice variability, both $I_{enr}$ and $Br_{enr}$ are calculated using Na as an indicator of sea salt content in the samples. The possibility of an IPO-related signal, transmitted through Na concentrations, cannot be discounted from contributing to the apparent correlation of $I_{enr}$ and $Br_{enr}$ in DSS0506 core samples.

## 3.5 Bromine seasonality

Due to the availability of consistent and highly resolved data from the DSS1213 core, it is possible to investigate seasonal distributions of sodium and bromine at Law Dome. The seasonal cycle of major and minor ions has been previously reported for Law Dome (Curran et al., 1998) but for bromine only four years are available (Spolaor et al., 2014). Here we present the DSS1213 record, spanning the period 1987 to 2012. The seasonality of sodium and $Br_{enr}$ are shown in Fig. 7. Sodium shows a broad period of high concentrations from April to September, with lower concentrations during the summer. This pattern broadly agrees with that reported by Curran et al., (1998) although those authors found a sharper sodium peak in late winter, associated with both the local sea ice maximum and the strongest local wind fields. We also note that the highest concentration of sodium is found in March-April and is likely due to one of '*a small number of large storm events which occasionally occur early in the year, lofting higher concentrations of sea-salt aerosol onto the summit of the dome*' (Curran et al., 1998).

The trend observed for $Br_{enr}$ is one of lower values in winter and higher values from November to February. There is high variability in the sodium signal throughout the year, whereas the $Br_{enr}$ signal is most variable during the summer months. The seasonality of $Br_{enr}$ found here confirms that suggested by the four-year data series presented by Spolaor et al. (2014), with a broad summer peak in $Br_{enr}$. Satellite observations of atmospheric BrO in polar regions suggest an early spring peak in bromine activity in Antarctica (Spolaor et al., 2014), thereby implying that additional processes may be occurring in the snowpack during the summer after the peak atmospheric bromine explosion has occurred. While snowpack remobilisation at Law Dome is minimal, it might be the case that photochemically-driven heterogenous recycling of bromine occurs in the snowpack after the springtime occurrence of the bromine explosion. This effect requires further investigation, from satellite and ground-based observations to weekly surface snow sampling, to be fully characterised and understood.

## 3.6 Bromine deposition across Law Dome

Variability in sodium and bromine was investigated along the transect line from Casey station to the Law Dome summit. The traverse route is indicated in Fig. 1, where 11 samples were collected over a period of 4 hours on the 10th February 2016. Conditions were clear with low wind and good visibility. Details of the sampling sites are included in Table S1 (supplementary material). The sodium and bromine results for each station are shown in Fig. 8.

Substantial variability can be seen along the traverse, with increasing fluxes of sodium and bromine approaching the Law Dome summit. It should be noted that the sampling was undertaken on the 'lee side' of Law Dome, across a region which generally exhibits low annual accumulation on the order of <200 kg m$^{-2}$. The final 25 km of the traverse sees a threefold-increase in annual accumulation to >600 kg m$^{-2}$. Higher fluxes of Na and Br in zones of higher accumulation suggest that both Na and Br are wet-deposited over the eastern half of Law Dome. The data presented here represent a first attempt to capture the surface variability of sodium and bromine across Law Dome. Future expeditions to Law Dome should incorporate a procedure for surface sampling along the traverse line and ideally further east toward the origin of the precipitation pathway over Law Dome.

## 4. Conclusions

The Law Dome site is ideal for studies of sea ice proxies, due to the regular and high level of annual precipitation allowing detailed studies of seasonality as well as the optimal preservation conditions. We find here that bromine enrichment over the past century displays some similarity to that of MSA, which has been previously used for reconstruction of local sea ice extent. The bromine-based reconstruction of the sea ice area in the 90110° E Antarctic sector suggests a reduction of sea ice area over the 20th century and hence is supportive of previous findings based on MSA. In agreement with satellite observations there is some indication of an increase in sea ice area since 1990.

In agreement with a previous study of halogen seasonality at Law Dome, we find that bromine enrichment displays regular seasonality with a broad summer peak. Iodine enrichment appears to be correlated to bromine enrichment on decadal scales but not annual scales, suggesting that despite their different chemical processes of emission and deposition, a multidecadal climate signal such as the Interdecadal Pacific Oscillation may act as a common influence on the enrichment of both halogen elements.

More extensive sampling across Law Dome and further inland should be considered in future field seasons. Samples collected during a traverse from Casey station to Law Dome DSS site display a consistent increase in deposition fluxes from west to east, as expected from the easterly cyclonic systems responsible for the precipitation at Law Dome. Additional

traverse data should be collected at the next available opportunity and ideally, the traverse should be extended over the dome summit and onto the eastern side of the dome. The Law Dome halogens record presented here should be extended further back in time and similar studies should be undertaken at other Antarctic locations, to ensure consistency and validation of the sea ice reconstruction presented here.

**Data availability**

Data will be made publicly available online through the NOAA Paleoclimatology database and PANGAEA data repository.

**Acknowledgements**

The research leading to these results has received funding from the European Research Council under the European Union's Seventh Framework Programme (FP7/2007-2013) grant agreement #610055 "Ice2ice". The Australian Antarctic Division provided funding and logistical support for the DSS ice cores (AAS projects 4061 and 4062). G. Spreen has been supported by the Institutional Strategy of the University of Bremen, funded by the German Excellence Initiative. All correlations and associated confidence levels reported here have been calculated using the program R via the RStudio interface. We thank two anonymous reviewers for their helpful suggestions and constructive comments.

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

Table 1. Statistical summary of time series data presented from Law Dome Dome Summit South (DSS) ice cores.

| Sample designation | Time interval | No. years | Sodium (ppb) | | | Bromine (ppb) | | | Ln(Brconc) | | Ln (Brenr) | |
|---|---|---|---|---|---|---|---|---|---|---|---|---|
| | | | mean | median | variance | mean | median | variance | mean | variance | mean | variance |
| DSS1516 snowpit | 2015-2016 | 0.8 | 241 | 179 | 91078 | 1.4 | 0.8 | 3.5 | -0.34 | 1.5 | 0.1 | 4.6 |
| DSS1213 core | 1987-2012 | 25 | 115 | 106 | 4355 | 2 | | 0.2 | 0.67 | 0.06 | 1.8 | 0.1 |
| DSS0506 core | 1927-1986 | 59 | 77 | 74 | 571 | 3.3 | | 9.7 | 0.88 | 0.57 | 2.5 | 0.6 |

Table 2. Correlations between bromine and iodine enrichments and First year Sea Ice (FYSI) areas calculated for two sectors adjacent to Law Dome. Correlations that are significant at the 99% level or above are shown in bold.

| Ln(Br$_{enr}$) | # years | FYSI 90-110 °E | | FYSI 110-130 °E | |
|---|---|---|---|---|---|
| | | $r^2$ | p-value | $r^2$ | p-value |
| Jan-Dec (summer-summer) | 35 | **0.357** | **<0.001** | 0.006 | ns |
| Jul-Jun (winter-winter) | 35 | 0.17 | <0.05 | **0.20** | **<0.01** |
| Sep-Nov (Spring only) | 35 | 0.18 | <0.05 | 0.02 | ns |

| I$_{enr}$ | # years | FYSI 90-110 °E | | FYSI 110-130 °E | |
|---|---|---|---|---|---|
| | | $r^2$ | p-value | $r^2$ | p-value |
| Jan-Dec (summer-summer) | 12 | 0.07 | ns | 0.24 | ns |
| Jul-Jun (winter-winter) | 12 | 0.02 | ns | 0.03 | ns |
| Sep-Nov (Spring only) | 12 | 0.09 | ns | 0.005 | ns |

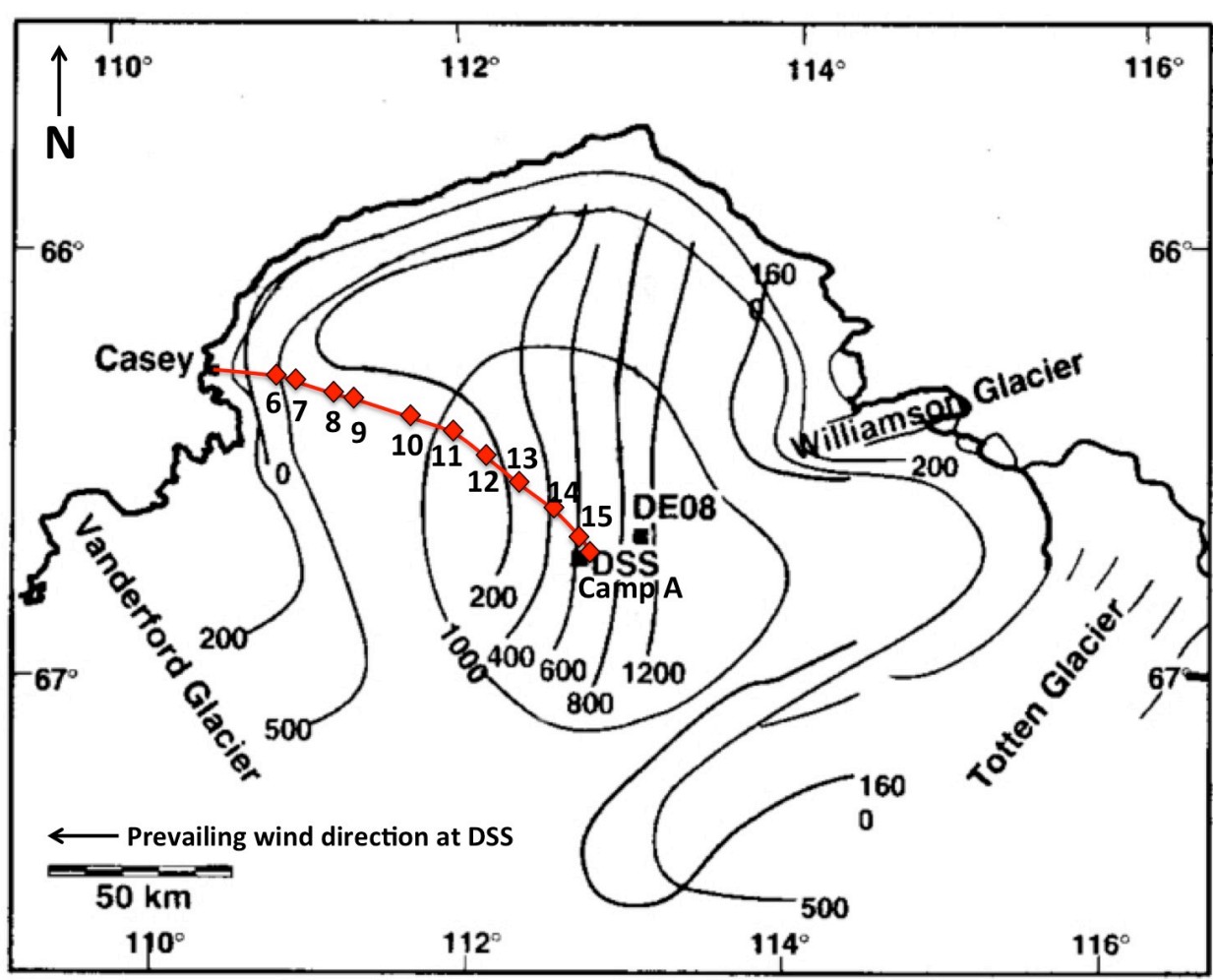

**Figure 1: Map of Law Dome with contours of elevation (m asl) and accumulation (kg m$^{-2}$ a$^{-1}$). The red line indicates the traverse route from Casey station to Camp A located near the Law Dome Summit. Red diamonds indicate the LDT surface snow sampling stations. DSS1516, DSS1213 and DSS0506 sampling sites are all in the vicinity of Camp A.**

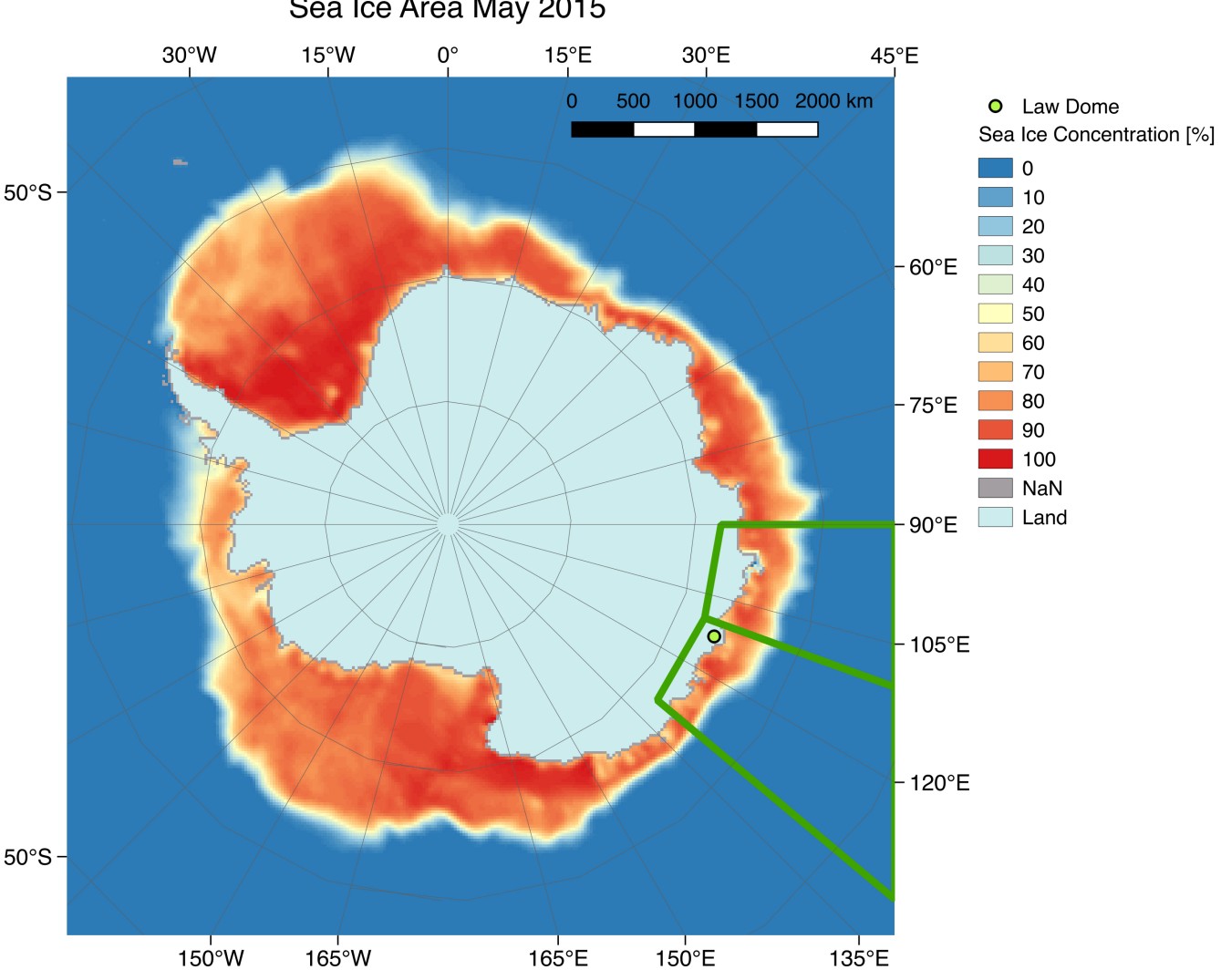

**Figure 2: Antarctic sectors used for evaluating sea ice trends. Two sectors (90-110° E and 110-130° E) adjacent to Law Dome have been isolated and used to calculate past sea ice area. The image shows and example of sea ice area for the month of May 2015 given as sea ice concentration (%) per grid cell.**

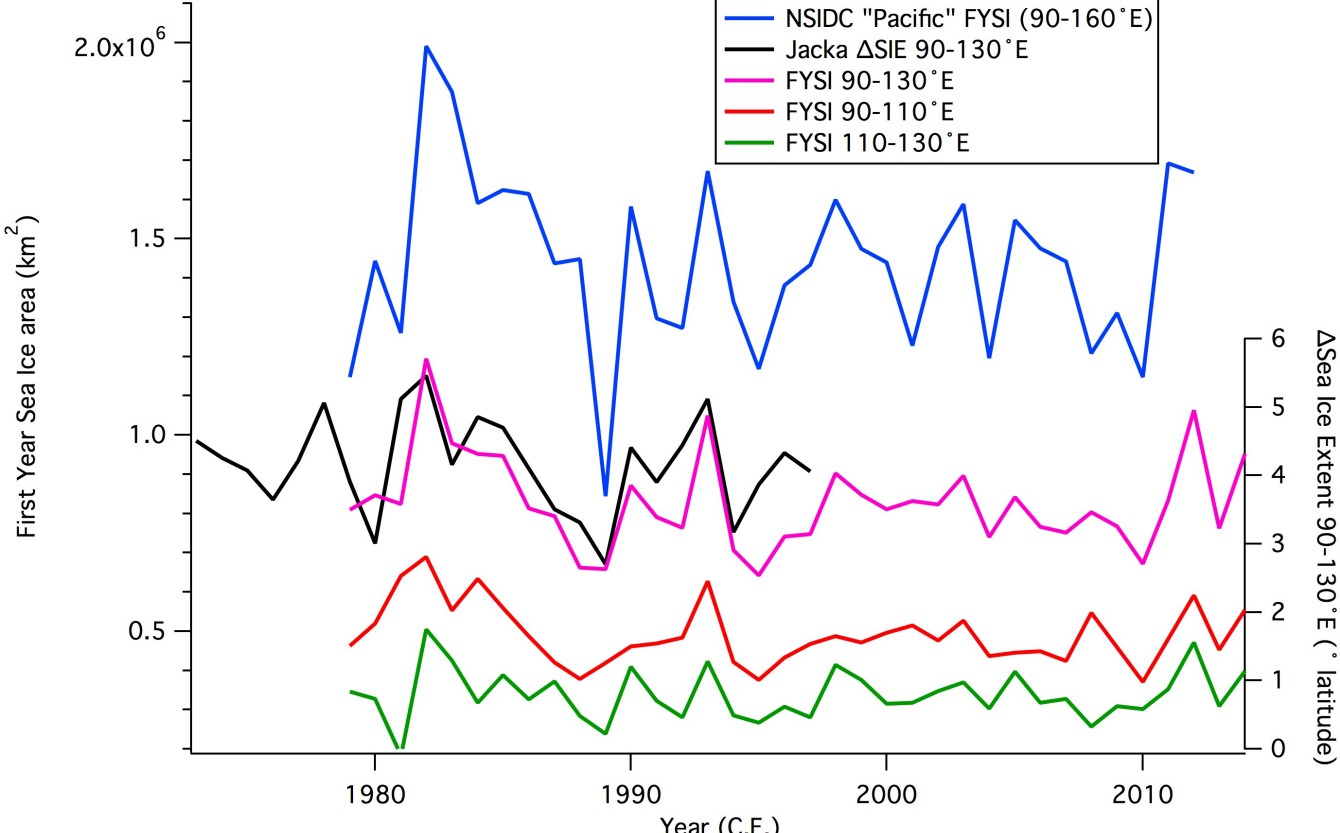

**Figure 3: First year sea ice (FYSI) areas calculated from satellite microwave radiometer observations. The 90-110° E and 110-130° E data shown here correspond to the two sectors indicated in Fig. 2. The 90-130° E FYSI data series (fuchsia) is the sum of the 90-110° E (red) and 110-130° E (green) data series'. For comparison to previous studies, we include corresponding data series from Jo Jacka (Jacka, 1998) and NSIDC (Fetterer et al., 2002). Note the NSIDC Pacific sector covers a larger sector (90-160ºE) than the**
10 **other records shown here.**

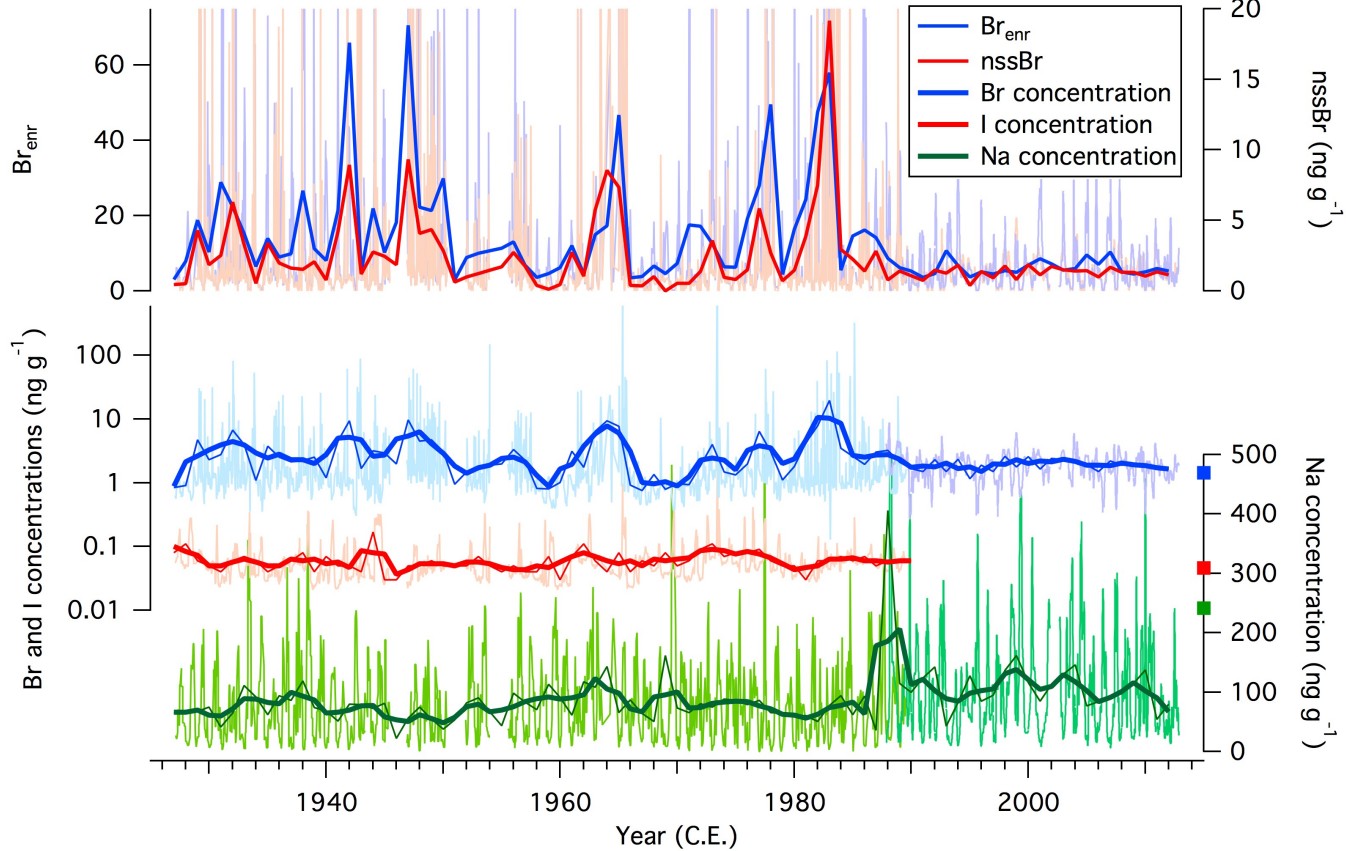

5  **Figure 4: Time series' of sodium, iodine and bromine concentrations as well as nssBr and Br$_{enr}$ at Law Dome. Raw data are shown in pale colours, with annual means shown by a thin line and three-year running means shown by a thick line. For nssBr and Br$_{enr}$, only raw data (pale colours) and annual averages (thick line) are shown. Different shades of blue and green are used to distinguish data from DSS0506 (1927-1989) and DSS1213 (1987-2013). The squares indicate average values from the DSS1516 snow pit.**

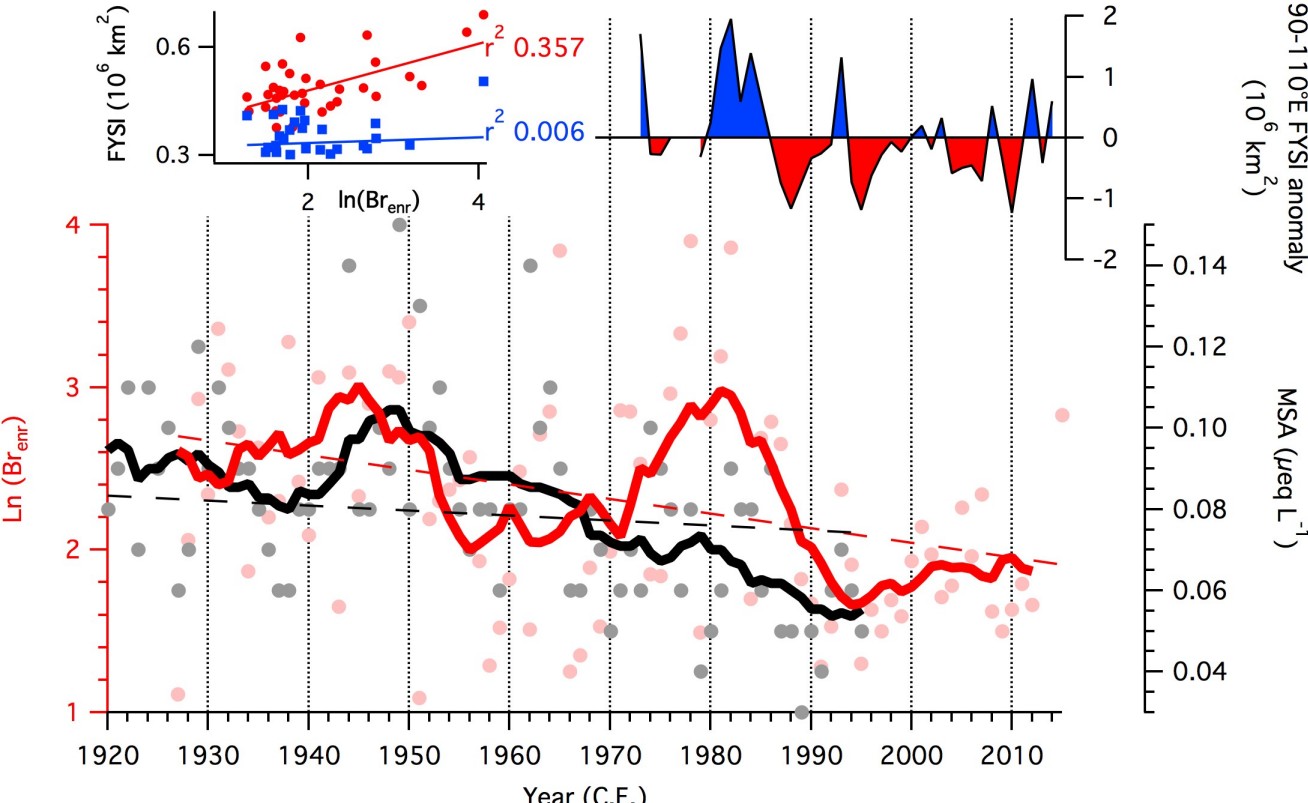

**Figure 5: Bromine enrichment (Br$_{enr}$, red), MSA (black) and First Year Sea Ice (FYSI) at Law Dome. Bromine enrichment and MSA data are shown as annual averages (circles) as well as 11-year (thick lines) running means. Linear regression trends are shown as dotted lines. FYSI areas in the 90-110° E sector (top right) are shown as annual anomalies from the 1973-2014 average. Shown in the top left panel are correlations between Br$_{enr}$ and FYSI areas in the 90-110ºE (red) and 110-130ºE (blue) sector.**

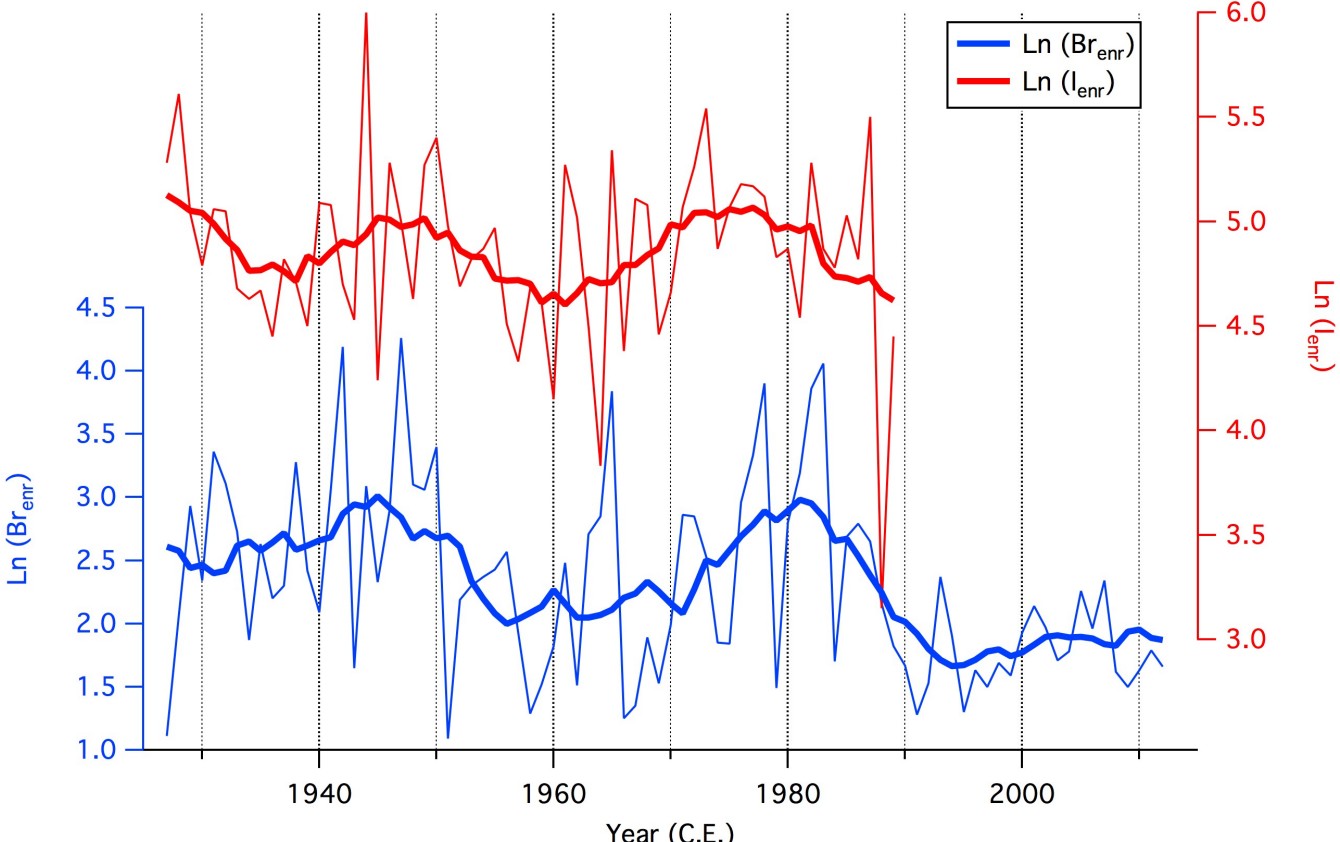

5 **Figure 6: Time series' of bromine and iodine enrichment beyond sea salt concentrations. As described in Sect. 3.2, sea salt is represented by sodium. Bromine and iodine show similar trends, pointing to a common source of variability.**

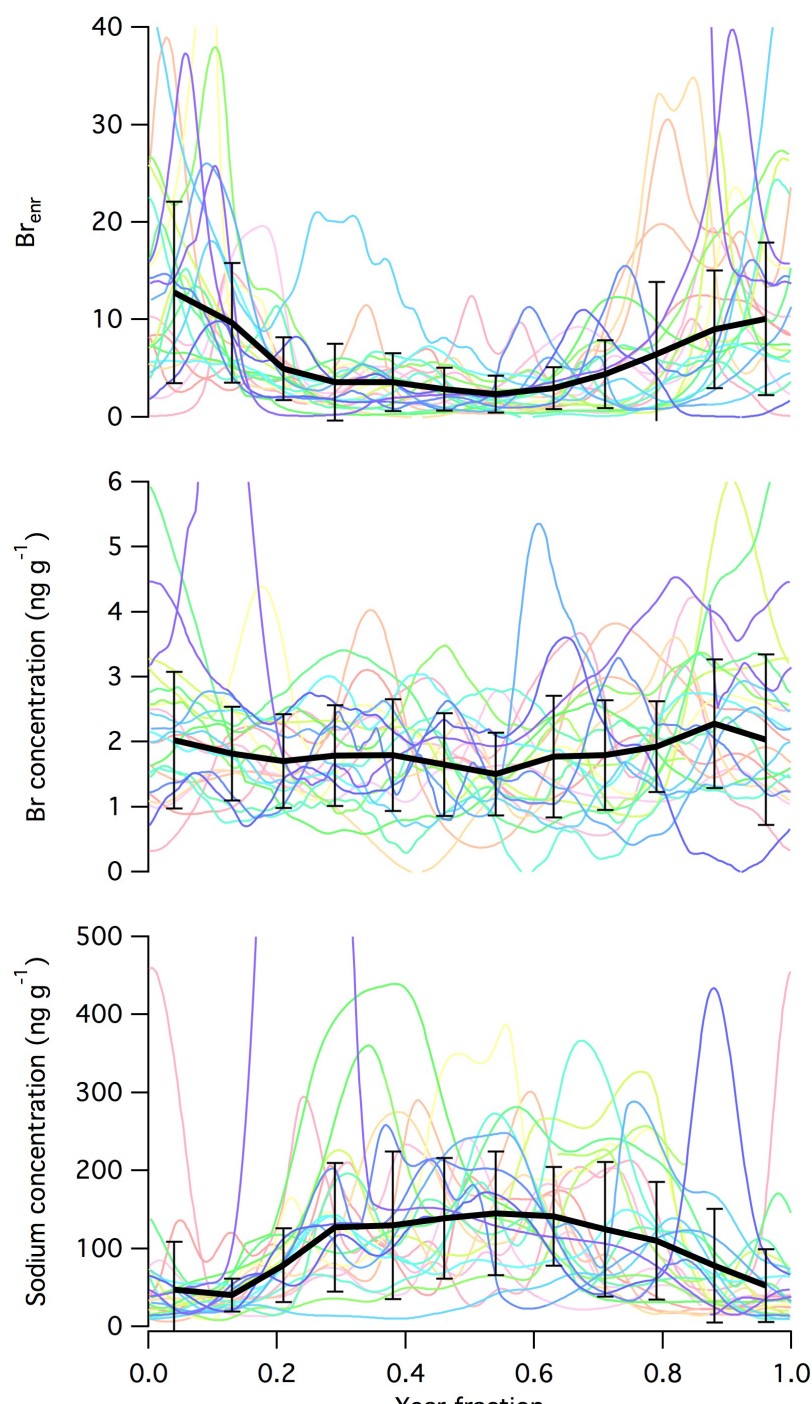

**Figure 7: Seasonality of sodium and bromine concentrations and bromine enrichment (Br$_{enr}$) in the DSS1213 firn core. Each colour corresponds to one year of data with the same year represented by the same colour. The year fraction has been separated into 12 months, with 0.42 representing January and 0.96 representing December. The error bars show 1 standard deviation for the 26 years sampled.**

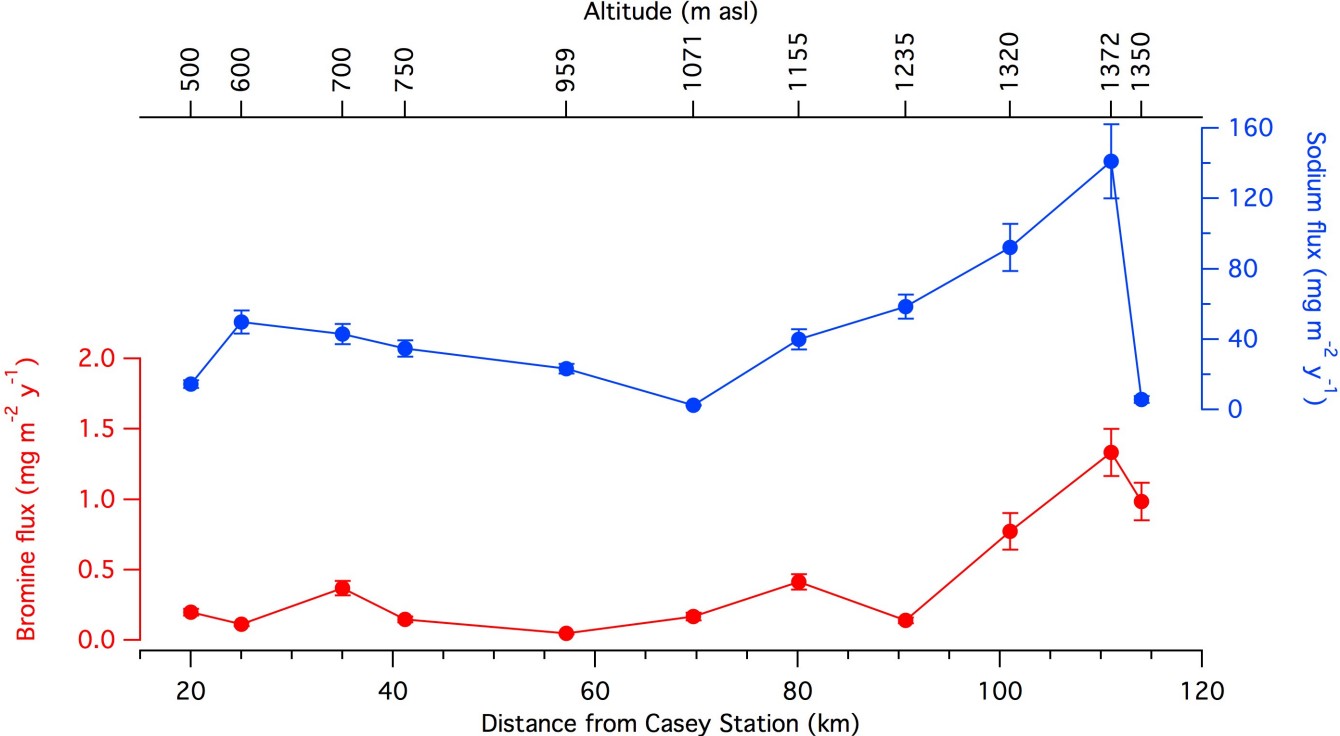

**Figure 8: Fluxes of sodium and bromine along the 2016 Law Dome Traverse (indicated in Fig. 1). Uncertainies include**
5  **uncertainties in the analysis (9% Br, 4% Na) and accumulation (10%).**