# Peer review of "Sea ice-related halogen enrichment at Law Dome, coastal East Antarctica"

_Climate of the Past, 2016_

## Referee Comment (RC1) · Anonymous Referee #1 · 21 Aug 2016

Vallelonga et al. present new halogen (Br and I) data from ice and snow samples at Law Dome, Antarctica. The manuscript is well-structured, clear and concise with nearly all the appropriate information provided. As in previous studies by several of the authors, the enrichment of Br (Brenr), relative to the Br/Na ratio of seawater is suggested as a proxy for first year sea ice. Although the concentration of Br itself shows no seasonality, Brenr is shown to peak in the spring-summer months and this is attributed the 'bromine explosion', a series of self-catalytic photochemical reactions understood to occur over sea ice in the springtime. Two ice cores extend the Brenr data set back to the early 20th century, and a gradual decline is seen, broadly similar to that observed for MSA (a relatively well-understood sea ice proxy) at Law Dome. A correlation between first year sea ice (FYSI) and Brenr is found, suggesting that Brenr could be a potential sea ice proxy. A set of surface snow samples from a traverse provide a first opportunity to

consider spatial variability in the Br/Na relationship around Law Dome.

As outlined by the authors, Law Dome is an obvious site for Antarctic sea ice proxy development. Its high accumulation rate, minimal multi-year sea ice and relatively simple meteorological conditions mean that the influence of complicating factors are reduced, plus the Curran et al. (2003) paper solidly established MSA as a proxy for local sea ice conditions. In this respect, this study is a timely and logical next step in the exploration of halogens as potential sea ice proxies. The study presents useful data that allow seasonal, inter-annual and century-long trends to be examined, greatly adding to the halogens data for this site. Some careful consideration of the interpretations and claims made is required, as I will detail below. In addition, the interesting surface snow traverse samples should be investigated further. This study should be suitable for publication in CP, providing the introduction and discussion are expanded to consider the complexities of halogen atmospheric chemistry and their potential impacts on the utility of Br and I as sea ice proxies.

Major comments The Introduction (3rd paragraph) needs expanding to set this study into context of previous halogens work on snow and ice samples. Br is not a "well-established" sea ice proxy and halogen atmospheric chemistry is highly complex. This needs to be made clear from the outset. It would also help to justify the need for sea ice proxy development at Law Dome, particularly the sample transect which is barely discussed at the moment. The questions surrounding halogen recycling, transport and deposition, aerosol vs. gas phase species are skipped over here but they need to be addressed. For example, in the early study quoted, Spolaor et al. (2013b) describe a mechanism by which Br is depleted relative to Na in glacial periods. They propose that Br is enriched over the sea ice but depleted inland (at Talos Dome) because the sea ice is further away in glacials and all the gaseous-phase HBr is deposited en-route. However, in a later paper, in the Arctic this time, Spolaor et al. (2016, Cryosphere) show good correlations between sea ice area and Br - the sign of correlation has changed to positive. The transect samples could be used to directly address the issue of whether

Br is transport/deposited as gaseous HBr or sea salt aerosol (see Simpson et al. 2005, GRL).

In the Spolaor et al. (2016) paper Br enrichment and Br excess (which is nssBr as I understand it) are plotted (Fig. 6). Could the same be done here? This would rule out a scenario in which the sea salt input (of Na and Br) changed but the speculated Br explosion component stayed constant, in which case Br enrichment would change but nssBr would remain the same. Along the same lines, if the similar multi-decadal variability in Br and I enrichment results from meterological/transport-related modification of the sea salt loading (IPO, Vance et al., 2015, 2016) as speculated, nssBr should look different to Brenr (maybe no change?).

The attraction of Law Dome for this study is the MSA-sea ice relationship established by Curran et al. 2000. Unfortunately, the similarity between MSA and Brenr ends at them both showing a slight decrease over time. Statements on pg 10 3rd paragraph and in the conclusion should be scaled back. It seems at least equally likely that the multi-decadal variability in Br enrichment is related to meteorology (as pg, 11, line 10) and not sea ice.

The significant, but rather weak, correlation between ln(Br enr) and FYSI then becomes central to the study and the relationship is not obvious from Fig.5. This deserves a separate figure or sub-figure.

All data (including raw Br, I and Na) should be made available in supplement or online database as indicated.

Minor comments - O'Dwyer et al., 2000 study Pg. 3 line7 is focused on the Arctic, not an Antarctic MSA record. - Sect 2.2, pg. 6 line 4: All data presented should be accompanied by data quality information (blanks, precision, accuracy). Representative 2-sigma uncertainty bars should ideally be included on figures (if they are large enough to see!). Please add this information to the supplement and include some discussion if it helps to resolve the differences between the two cores (Fig. 4). Were replicate samples

measured in both labs? - Sect. 3.1, pg. 8 line ∼15: If the introduction is improved, it may not be necessary, but it should be made clear here that both Na and Br come from the sea salts (from open ocean and sea ice surface), but that Br levels in atmosphere can be "enriched" through the Br explosion. - Sect 3.1 pg. 8, line ∼25: This section is confusing. Clearly Br is less variable in the latter part of the record (DSS1213) and this is attributed to smoothing in melter system, which I can understand. However, the Na record shows greater variability or may even show a step-wise increase (?). Why is the Na record not smoothed like Br? What does that suggest about the annual cycles of the two species and the seasonal Brenr calculated from them? - Sect. 3.2 Pg. 6 line 5: I'm not sure I understand the justification for using natural log, as it relates to Br chemistry. Br enrichment is a ratio, not a concentration. Please clarify. - Sect 3.2, pg. 9 line 26: Levine et al. (2014) do not discuss post-depositional remobilization, please add another citation here. line 27: do you mean "seasonality" of sea salt deposition? line 30: How is the correlation "consistent", with what? - Sect 3.4, pg 10, line 30: Do you mean iodine enrichment in snow, not sea ice? - Sect 3.4, pg 10, line 31: Do heterogeneous reactions really "release" particulate species? - Sect. 3.4 Why is the seasonality of iodine not shown on Fig. 7? Does the correlation stated refer the results from this study or previous one? Later in sect.4 the text says that "iodine enrichment displays broad summer peak" but that is not shown anywhere is it? - Sect 3.5, pg11 line 31: Brenr is not a concentration. - Sect 4, pg. 13. paragraph 2: this paragraph makes little sense because no iodine seasonality data is presented. I also struggle to see how iodine can experience greater "meteorological disturbance" compared to Br. Differences in chemical reactions in snowpack and atmosphere seem more likely. - Figure 1: Could an arrow be added to show direction of cyclone movement across site? Also a north arrow? Please add explanation of ice movement arrows to caption, or remove them. - Figure 2: This would be easier to interpret quickly if axes were degrees, rather than arbitrary grid. Is the blue color meaningful, probably outlining the two regions is enough? - Not sure that I see the worth of Figure 3 – seems like a technical detail. Suggest moving to the supplement. Some of the technical detail in Sect. 2.4

could also be moved to the supplement. - Figure 4: Why are I concentration data from DSS1213 not plotted on Fig. 4 or enrichment on Fig. 6)? Methods indicate that iodine was measured.

Technical notes - abstract, line 23. Change "particularly" to specifically – correlation only found for the one sector. - line 23: Iodate should be iodate

---

## Referee Comment (RC2) · Anonymous Referee #2 · 10 Oct 2016

Comments to the paper: Sea ice –related halogen enrichment at Law Dome, coastal East Antarctica P. Vallelonga et al.

General Comments

The paper is concerning the possibility that Br-enr and I-enr can be used as proxy markers for sea-ice extension and/or persistence in Antarctica. The topic is interesting, especially for paleo-climate studies concerning the reconstruction of sea-ice dynamics from chemical stratigraphies of ice cores. Besides, every new information of the chemistry of halogen compounds on snow and sea-ice surface is interesting in order to understand their relationship with marine biological activity, tropospheric ozone and photochemical processes at the sea/atmosphere interface. However, in my opinion, the majority of the information about concentration in the snow (and firn and ice) and

seasonal trends of Br and I was already reported in previous papers and, especially, the relationship between Br-enr and I-enr with sea-ice dynamics (the main goal of the paper) seems to be not sufficiently strong as the Authors assessed. Some interpretations of the temporal trends, especially those concerning the comparison with first year sea ice (FYSI) data, are, in my opinion, not fully corresponding to the profiles shown by the plots (see specific comments). Besides, some improvement should be made in the methodological sections and in the data discussion. In conclusion, I think that the manuscript is not ready, in this form, to be accepted for publication on Climate of the Past journal. However, since the topic is very interesting and a huge analytical work was made to analyze snow, firn and ice samples, I'd like to encourage the Authors to submit an improved manuscript, possibly taking into account my criticisms and suggestions.

Specific and minor comments

Lines 5-8, page 2. Authors should give some summarized information on the chemical processes involved in the "photochemical recycling above salt-rich snow and ice surfaces", even if a reference is correctly cited.

Line 23, page 2.Authors are requested to indicate the DL for bromate and iodide in order to have an idea about their possible (maximum) concentration levels in the Talos Dome samples.

Lines 28-29, page 3. Are the snowfalls really so "regular" to provide very detailed (month-by-month?) stratigraphies? For the time covered by the DSS1213 firn core, some basic information about the snowfalls frequency could be given in the Section 2.

Lines 18-19, page 4. DSS0506 sub-samples were melted and refrozen in 2006 and analyzed in 2014. Have the Authors some evidences about possible effects of melting/refreezing cycle and long-time storing on the determination of Br and I?

Section 2.2 – Analytical measurements. Even if sufficient references were cited, some

analytical methods performances should be here summarized (reproducibility, accuracy, DL, blank values). Was an inter-calibration exercise made between Australian and Italian laboratories?

Line 30, page 5 and line 3, page 6. Mass resolution (m/dm) is dimensionless. Please, delete "amu".

Lines 4-5, page 6. Please, summarize the method performances in terms of accuracy and difference between blanks and samples values.

Line 9, page 6. The term "core" has to be referred to DSS0505 and not to DSS1516. Where the DSS1516 snow pit were analyzed? At lines 18-19, page 5, Authors report that DSS1516 snow pit samples were analyzed for Br in Australia. Maybe, in Italy the same samples (or samples from a parallel column in the snow pit) were analyzed even for I and Na, other than Br. In this case, were the Br values compared?

Section 2.3. Which samples were analyzed for IC? Were the Na values reported in the manuscript analyzed by IC (soluble fraction) or by ICP-MS (probably total content)? The analyzed Na fraction could play a not-negligible role in evaluating the Br-enr and I-enr fractions, if Na, and not ssNa, is used as sea spray marker. Also for IC measurements, the methods performances (at least for Na and MSA) should be here summarized.

Lines 2-3, page 7. Please, reword the sentence.

Lines 7-8, page 8. I think that median is more suitable than geometric mean in evaluating the asymmetry of the data sets, by comparison with the mean values.

Line 11, page 8 and following. The calculation of nssBr, nssI, Br-enr and I-enr have to be made by using ssNa, and not total Na, as sea spray marker. I'm aware that, in a coastal site, the majority of the Na content in the snow is originated by sea spray, but also local or long-range dust could give not negligible contributions, at least in particular transport events. As well known, the nssNa fraction (and then ssNa by difference)

can be easily evaluated by using Al (if Na is measured by ICP-MS) or nssCa (if Na is measured by IC) as crustal markers and knowing the Al/Na or Ca/Na ratios in the uppermost Earth crust.

Line 12, page 8. Please, change "sodium" in "ssNa".

Line 15, page 8. Authors are requested adding a short description of meaning and seasonal occurrence of the "bromine explosion" events.

Lines 17-18, page 8. The calculation of the enrichment factors of different snow and aerosol components with respect to seawater composition is well established and cannot be attributed to some of the Authors.

Line 18, page 8. Please, change "Na" in "ssNa".

Line 22, page 8 and following. In my opinion, the tentative explanation of the different variability of Na and Br in the two records (DSS0506 and DSS1213 firn cores) appears to be not convincing. Table 1 and Figure 4 show that DSS1213 Na profile has higher mean values and much higher variability with respect to the DSS0506 record. On the contrary, the DSS1213 Br profile shows a very sharp smoothing of the 3-yr running mean and lower mean values, with respect to the DSS0506 record. A so large Na and Br opposite variability cannot be attributed, in my opinion, to selective (Br, with respect Na) "memory effects" or to a different depth resolution of the analytical methods (melter vs discrete samples). Memory effects are usually related to the matrix and not to single components; besides they could play a smoothing effect (but not as large as for the Br) and cannot increase the variability (as shown by the Na profile). Differences in measurements resolution (continuous melting vs discrete samples) are fully able to change the data variability, but not in opposite sign for the two components; besides, the different resolution (if not too much large with respect to accumulation rate) should be not able to change the 3-yr mean profiles. Authors are requested to report the estimated depth resolution for the continuous melter system. Finally, snow pit data show very higher Na values and similar Br concentrations (it is difficult to evaluate little

differences in a logarithmic scale; Authors are requested to add the snow pit mean values in Table 1), with respect to DSS1213 firn core. How the Authors can explain these patterns? Could the different profiles be caused by different analytical methods in the different laboratories?

Line 6, page 9. I cannot understand how the variability in the Br-enr data "may act to artificially increase the correlation". Usually, higher variability could cause a loss of correlation or make more difficult the evaluation of a possible correlation between two parameters. Authors are requested to explain their thought.

Line 17, page 9. The Br-enr – FYSI correlation is poor (max $R^2$ value: 0.32) for the 90-110 °E sector and null for the 110-130 °E sector. This sector selectivity seems to be too large (see, also, I-enr, that shows a completely opposite pattern) and could imply that the correlation between the two parameters (Br-enr or I-enr with FYSI) is weak and possibly covered by other factors, such as atmospheric circulation modes (with an opposite effect for Br-enr and I-enr?). Indeed, a 0.32 $R^2$ value, even if significant at the 99% level, means that just 1/3 of the Br-enr variance can be attributed to changes in FYSI. In my opinion, the $R^2$ values are not sufficient to support the Authors hypothesis (see also my below comments to the Figures 5 and 6).

Line 20, page 9. The MSA-FYSI correlation in the 80-140 °E sector cannot support the Br-enr – FYSI correlation because the last correlation is highly sector selective and the 80-140 °E sector covers a sector (110-130 °E) in which the Br-enr – FYSI correlation is completely absent.

Lines 24-30, page 9. As before discussed (my comments to line 17, page 9), Authors attribute to several possible "noise effects" the Br-enr – FYSI correlation variability as a function of summer-summer or winter-winter intervals. In my opinion, the correlation is always poor and $R^2$ values depend on too much factors to be confident. The last sentence (lines 29-30) is not supported by the data.

Lines 14-15, page 10. These limitations in the comparison of MSA and Br-enr temporal

trends are correct, but we have to consider that we are comparing 11-yr mean profiles. Therefore, some limiting factors, especially the different seasonal pattern, surely play a minor or null role.

Line 21 and line 23, page 10. I cannot see in Figure 5 a significant increase, with respect to the noisy baseline along the multi-decadal trend, of MSA during the periods 1920-30 and 1975-85. For instance, the MSA profile in the period 1955-67 could show similar positive anomalies. The only significant increase is related to the period 1940-55.

Line 22, page 10. I agree that MSA and Br-enr profiles show a common multi-decadal trend (a slight decrease), but the single common feature, around 1940-1955, shows peaks shifted of about 4 years. Besides, the trend in the period 1955-80 is opposite. Even neglecting the large 1970-90 Br-enr peak, which is not evident in MSA profile, the 1955-70 trends of Br-enr (increasing) is opposite to that of MSA (decreasing). I think that the agreement between the two parameters is weak and Authors should try to explain the observed differences in the temporal shift of the 1940-50 peaks and in the 1955-70 opposite trends.

Lines 25-26, page 10. I have some perplexities also concerning the comparison between FYSI and Br-enr. Unfortunately, the period covered by satellite measurements is short and a reliable comparison is difficult. However, I can see two clear evidences that should be explained. By observing the Br-enr large peak around 1970-90, we have to note that while it is correct that the highest value is synchronous with the 1982 large positive anomaly in the FYSI, its temporal evolution does not follow the FYSI dynamics. Indeed, the Br-enr peak show an abrupt increase when FYSI positive anomalies are not marked (unfortunately, FYSI 1977 and 1978 satellite data are missing, but 1976 and 1979-80 data show null or slightly negative anomalies). Besides, after the 1981 peak, Br-enr quickly decreases, while FYSI shows relevant positive anomalies until 1985. Finally, almost continuous FYSI negative anomalies from 1993 to 2010 do not cause negative peaks in the Br-enr profile that, on the contrary, shows a continuous

and clear increase. I think the Authors should reconsider their assessments and better discuss (and, possibly, interpret) the complex FYSI-Br-enr relationship. As the Authors report in the next section, the two Br-enr peaks (around 1945 and 1981) could be related to changes in atmospheric circulation modes (e.g., changes in the IPO) that could include (but not only) changes in sea ice dynamics.

Lines 6-7, page 11. The correlation coefficient between I-enr and FYSI for summer-summer 110-130 °E sector ($R2 = 0.42$) is higher than that of Br-enr for 90-110 °E ($R2 = 0.32$), even if the p-value is slightly lower (<0.01, with respect to <0.001, but anyway significant). Authors should discuss why the correlation between I-enr and FYSI is so sector selective, and opposite of that, similarly sector selective, between Br-enr and FYSI.

Line 10, page 11. A correlation with $R2 = 0.27$, even if the value is statistically significant, is really too poor and cannot demonstrate that the two parameters are correlated (just $\frac{1}{4}$ of the variance of a parameter is explained by the variability of the other). However, by observing figure 6, the two profiles are very similar. Maybe, the correlation is poor because there are temporal shifts between the peaks of the two records. Indeed, the 1945 Br-enr peak leads the I-enr peak and the opposite pattern is visible for the 1980 Br-enr peak. In my opinion, the Br-enr – I-enr relationship deserves an improved discussion.

Lines 13-18, page 11. The relationship of I-enr and Br-enr with IPO is potentially very interesting. Unfortunately, Authors barely touches on the topic. The Authors should improve the discussion and evaluate how the IPO changes can affect the Br and I emissions or transport processes. For instance, why positive-to-negative and negative-to-positive IPO phase changes cause the same effects on I-enr and Br-enr profiles? Which are the relationships between IPO phases and atmospheric circulation around Antarctica or sea-ice dynamics?

Section 3.5. Maybe this section should be moved just after (or inside) section 3.1.

No novelty on the seasonal pattern of Br is here reported, with respect to previous results on shorter data series. The tricky dephasing between spring Br explosion and Br summer maximum in the snow is not explained (and I agree that this pattern has to be in deep studied).

Line 16, page 12. Figure 8 show fluxes and not concentrations. Na fluxes are very higher in the final 25 km, with respect to more coastal sites.

Line 20, page 12. This Na and Br pattern is interesting and should be enlightened. Higher fluxes in higher snow-accumulation sites mean that Na and Br deposition occurs mainly by wet-deposition, while dry deposition could be negligible. This fact can have implications in ice-core studies.

Conclusions. This section should be revised accordingly to the suggested manuscript changes.

---

## Author Comment (AC1) · 10 Nov 2016

CPD Law Dome Halogens Responses to reviewers

Anonymous Referee #1

We thank the reviewer for their interest in the article and their detailed comments. We would like to point out that since the initial manuscript submission, we have identified a bug in the code used to generate the sea ice areas (FYSI was overestimated), which has led to some changes in Figure 3 and some of the correlations in Table 2. Also, revision of the data has identified gaps in the ESMR 1973-1977 satellite dataset, so some years have been removed from the FYSI dataset as a result. The essential findings of the manuscript have not been changed by this new data.

Vallelonga et al. present new halogen (Br and I) data from ice and snow samples at Law Dome, Antarctica. The manuscript is well-structured, clear and concise with nearly all the appropriate information provided. As in previous studies by several of the authors, the enrichment of Br (Brenr), relative to the Br/Na ratio of seawater is suggested as a proxy for first year sea ice. Although the concentration of Br itself shows no seasonality, Brenr is shown to peak in the spring-summer months and this is attributed the 'bromine explosion', a series of self-catalytic photochemical reactions understood to occur over sea ice in the springtime. Two ice cores extend the Brenr data set back to the early 20th century, and a gradual decline is seen, broadly similar to that observed for MSA (a relatively well-understood sea ice proxy) at Law Dome. A correlation between first year sea ice (FYSI) and Brenr is found, suggesting that Brenr could be a potential sea ice proxy. A set of surface snow samples from a traverse provide a first opportunity to consider spatial variability in the Br/Na relationship around Law Dome.

As outlined by the authors, Law Dome is an obvious site for Antarctic sea ice proxy de- velopment. Its high accumulation rate, minimal multi-year sea ice and relatively simple meteorological conditions mean that the influence of complicating factors are reduced, plus the Curran et al. (2003) paper solidly established MSA as a proxy for local sea ice conditions. In this respect, this study is a timely and logical next step in the explo- ration of halogens as potential sea ice proxies. The study presents useful data that allow seasonal, inter-annual and century-long trends to be examined, greatly adding to the halogens data for this site. Some careful consideration of the interpretations and claims made is required, as I will detail below. In addition, the interesting surface snow traverse samples should be investigated further. This study should be suitable for publication in CP, providing the introduction and discussion are expanded to consider the complexities of halogen atmospheric chemistry and their potential impacts on the utility of Br and I as sea ice proxies.

Major comments

The Introduction (3rd paragraph) needs expanding to set this study into context of previous halogens work on snow and ice samples. Br is not a "wellestablished" sea ice proxy and halogen atmospheric chemistry is highly complex. This needs to be made clear from the outset. It would also help to justify the need for sea ice proxy development at Law Dome, particularly the sample transect which is barely discussed at the moment. The questions surrounding halogen recycling, transport and deposition, aerosol vs. gas phase species are skipped over here but they need to be addressed. For example, in the early study quoted, Spolaor et al. (2013b) describe a mechanism by which Br is depleted relative to Na in glacial periods. They propose that Br is enriched over the sea ice but depleted inland (at Talos Dome) because the sea ice is further away in glacials and all the gaseous-phase HBr is deposited en-route. However, in a later paper, in the Arctic this time, Spolaor et al. (2016, Cryosphere) show good correlations between sea ice area and Br - the sign of correlation has changed to positive. The transect samples could be used to directly address the issue of whether Br is transport/deposited as gaseous HBr or sea salt serveral (see Simpson et al

Br is transport/deposited as gaseous HBr or sea salt aerosol (see Simpson et al. 2005, GRL).

We have expanded the introduction to provide additional information regarding halogen chemical processes relevant to polar ice sheets. Changes have been made to the first paragraph of the introduction noting the complexity of halogen chemistry, and elaborating on the key uncertainties and processes relevant to this work. Additionally, examples of insitu observations and combined observations-model exercises have been cited.

Due to the quite different temporal scales involved, we consider it inappropriate to directly compare the findings of the 200 kyr Talos Dome record (2013, ACP) with those from the 50 yr Severnaya Zemlya record (2016, Cryosphere). Firstly we note that in the Severnaya Zemlya record, similar to Law Dome, we are dealing with a period of relatively well-constrained FYSI variability.

When considering glacial-interglacial changes such as those investigated at Talos Dome, changes in MYSI and FYSI areas are of substantially greater magnitudes and hence it is possible for the sampling site (ice core site) to change from a location of bromine enrichment to one of bromine depletion. In this light, the findings from Talos Dome are much more consistent with a recently published article reporting 120 kyr Br record from the NEEM ice core (2016, Scientific Reports) - both sites demonstrate substantial changes in Brenr linked to corresponding changes in sea ice extent in glacial and interglacial climates.

In the Spolaor et al. (2016) paper Br enrichment and Br excess (which is nssBr as I understand it) are plotted (Fig. 6). Could the same be done here? This would rule out a scenario in which the sea salt input (of Na and Br) changed but the speculated Br explosion component stayed constant, in which case Br enrichment would change but nssBr would remain the same. Along the same lines, if the similar multi-decadal variability in Br and I enrichment results from meterological/transport-related modification of the sea salt loading (IPO, Vance et al., 2015, 2016) as speculated, nssBr should look different to Brenr (maybe no change?).

The reviewer is correct that Br excess is identical to nssBr and in future we will consistently use nssBr. We have expanded Figure 4 to include both  $Br_{enr}$  and nssBr. Overall, there is good agreement between the two measures, as was also the case in the 2016 paper cited by the reviewer. This is because  $Br_{enr}$  and nssBr are essentially the same measure, represented in different ways. The only difference between  $Br_{enr}$  and nssBr is that one ( $Br_{enr}$ ) determines the relative difference while the other (nssBr) determines the absolute difference between Br found in the sample and Br expected from sea salt. Otherwise they are calculated from the same measurements of Br and Na and assume the same Br/Na seawater ratio. As a result,  $Br_{enr}$  is never less than zero, with values greater than 1 indicated enrichment of Br above sea salt levels and values less than 1 indicating depletion of Br with respect to sea salt levels (as was found for glacial Talos Dome samples).

Regarding the possible influence of IPO, it is important to recognize that IPO is not significant at daily timescales relevant to meteorological/transport processes. IPO is a low frequency (multi-decadal) mode of variability related to ENSO and indeed resembles smoothed (low frequency) ENSO variability. Both the IPO and ENSO have a low frequency impact on multidecadal variability across and beyond the Pacific Basin. Therefore while IPO is likely to influence sea salt levels in Law Dome (among many other locations), we do not expect IPO to influence the processes underlying Bromine explosion events. A more detailed investigation of IPO influences to Law Dome is presented by Vance et al. (GRL, 2015).

The attraction of Law Dome for this study is the MSA-sea ice relationship established by Curran et al. 2000. Unfortunately, the similarity between MSA and Brenr ends at them both showing a slight decrease over time. Statements on pg 10 3rd paragraph and in the conclusion should be scaled back. It seems at least equally likely that the multi-decadal variability in Br enrichment is related to meteorology (as pg, 11, line 10) and not sea ice.

We reduced have scaled back the statements regarding similarities between the MSA and  $Br_{enr}$  records and added a consideration of the potential influence of IPO on the records. Accordingly changes have been made to pg 10 (paragraph 3) and in the conclusions.

The significant, but rather weak, correlation between ln(Br enr) and FYSI then becomes central to the study and the relationship is not obvious from Fig.5. This deserves a separate figure or sub-figure.

An inset figure has been added to figure 5 showing the correlation between  $ln(Br_{enr})$  and FYSI for sectors 90-110°E and 110-130°E.

Updated Figure 5.

All data (including raw Br, I and Na) should be made available in supplement or online database as indicated.

**We have prepared the data files and these have been sent to the NOAA and PANGAEA paleoclimate databases for archiving.**

**Minor comments**

O'Dwyer et al., 2000 study Pg. 3 line7 is focused on the Arctic, not an Antarctic MSA record.

**The reference has been removed**

Sect 2.2, pg. 6 line 4: All data presented should be accompanied by data quality information (blanks, precision, accuracy). Representative 2-sigma uncertainty bars should ideally be included on figures (if they are large enough to see!). Please add this information to the supplement and include some discussion if it helps to resolve the differences between the two cores (Fig. 4). Were replicate samples measured in both labs?

The relevant data quality information for the Italian and Australian ICP-MS laboratories have been added. Figure 8 has been revised to include measurement and accumulation uncertainties. For the other figures, uncertainty bars are too small to be added (Figs 5, 6) or the resolution of the data shown precludes the addition of error bars (Figs 4, 7). Finally, a paragraph has been added to section 2.2.2 describing interlaboratory reproducibility measurements:

The reproducibility of measurements between the two laboratories was tested by analyzing 140 Greenland snow pit samples in both laboratories. Compatibility of the measurements (Supplementary Figures S1 and S2) showed a regression line with R2>0.9 (n=140, p

---

## Author Comment (AC2) · 10 Nov 2016

CPD Law Dome Halogens
Responses to reviewers

Anonymous Referee #2

General Comments
The paper is concerning the possibility that Br-enr and I-enr can be used as proxy markers for sea-ice extension and/or persistence in Antarctica. The topic is interesting, especially for paleo-climate studies concerning the reconstruction of sea-ice dynamics from chemical stratigraphies of ice cores. Besides, every new information of the chemistry of halogen compounds on snow and sea-ice surface is interesting in order to understand their relationship with marine biological activity, tropospheric ozone and photochemical processes at the sea/atmosphere interface. However, in my opinion, the majority of the information about concentration in the snow (and firn and ice) and seasonal trends of Br and I was already reported in previous papers and, especially, the relationship between Br-enr and I-enr with sea-ice dynamics (the main goal of the paper) seems to be not sufficiently strong as the Authors assessed. Some interpretations of the temporal trends, especially those concerning the comparison with first year sea ice (FYSI) data, are, in my opinion, not fully corresponding to the profiles shown by the plots (see specific comments). Besides, some improvement should be made in the methodological sections and in the data discussion. In conclusion, I think that the manuscript is not ready, in this form, to be accepted for publication on Climate of the Past journal. However, since the topic is very interesting and a huge analytical work was made to analyze snow, firn and ice samples, I'd like to encourage the Authors to submit an improved manuscript, possibly taking into account my criticisms and suggestions.

We thank the reviewer for their detailed comments and suggestions and hope that the revised manuscript is found to be suitable for publication. We would like to point out that the manuscript does present new information, especially with regard to Law Dome, where previously only a 4-year sequence of Br and I concentrations have been published. Additionally, we include measurements along a surface traverse from Casey station to Law Dome which, to our knowledge, are the first data regarding spatial distribution of halogens in snowpack. Many of the reviewer comments have already been addressed by our responses to reviewer 1.

We would also like to point out that since the initial manuscript submission, we have identified a bug in the code used to generate the sea ice areas (FYSI was overestimated), which has led to some changes in Figure 3 and some of the correlations in Table 2. Also, revision of the data has identified gaps in the ESMR 1973-1977 satellite dataset, so some years have been removed from the FYSI dataset as a result. The essential findings of the manuscript have not been changed by this new data.

Specific and minor comments

Lines 5-8, page 2. Authors should give some summarized information on the chemical processes involved in the "photochemical recycling above salt-rich snow and ice surfaces", even if a reference is correctly cited.

Additional information regarding photochemical recycling has been added:

Photochemical recycling of bromine primarily involves heterogeneous reactions of halide salts (such as HOBr and $BrONO_2$) in sea ice and snowpack leading to the emission of $Br_2$ molecules. $Br_2$ is then photodissociated into two $Br^-$ radicals that are available for further heterogeneous chemical recycling. Bromine explosion events primarily occur in early spring and summer, although winter sources of organohalide emissions have also been observed in coastal polar regions although the relative influence of such sources is still a topic of investigation (Impey et al., 1997; Nerentorp Mastromonaco et al., 2016; Simpson et al., 2007).

Line 23, page 2.Authors are requested to indicate the DL for bromate and iodide in order to have an idea about their possible (maximum) concentration levels in the Talos Dome samples.

The DLs have been added (38 pg $BrO_3^-$ $g^{-1}$, 7 pg $IO_3^-$ $g^{-1}$).

Lines 28-29, page 3. Are the snowfalls really so "regular" to provide very detailed (month-by-month?) stratigraphies? For the time covered by the DSS1213 firn core, some basic information about the snowfalls frequency could be given in the Section 2.

The Law Dome site is well known for its relatively plentiful and regular snowfall, as well as benign deposition conditions. A number of studies have investigated snowfall regularity and seasonality at the site (eg Morgan et al., 1997, J. Glacio.; McMorrow et al., 2004, Ann. Glacio.). These references are included in section 2.

Lines 18-19, page 4. DSS0506 sub-samples were melted and refrozen in 2006 and analyzed in 2014. Have the Authors some evidences about possible effects of melting/refreezing cycle and long-time storing on the determination of Br and I?

This work is the first direct comparison between long-term stored samples and freshly recovered core samples. We have two lines of information indicating that remelting does not influence the Br concentration as long as the sample is otherwise stored frozen in dark conditions:

Firstly, we note in section 3.1 (and Fig 4) the good agreement of Br concentrations in overlapping sections of DSS0506 (long-term storage and remelting) and DSS1213 (freshly sampled by continuous melting). Unfortunately I was not measured in the DSS1213 core so a similar comparison cannot be made at this time.

Secondly, we note that there is excellent agreement between Br measurements of Greenland snow pit samples (Supplementary Figure S1) measured in the two laboratories. Note that these samples were melted and refrozen three times (during sampling in Copenhagen, during analysis in Venice Italy, and during analysis in Perth Australia). The same samples in the same vials were melted three times and measured twice, futher indicating that Br is preserved as long as samples are kept frozen and stored in darkness.

[Figure]

Supplementary Figure S1

Section 2.2 – Analytical measurements. Even if sufficient references were cited, some analytical methods performances should be here summarized (reproducibility, accu- racy, DL, blank values). Was an inter-calibration exercise made between Australian and Italian laboratories?

Reviewer 1 had an identical comment and we copy here the response: The relevant data quality information for the Italian and Australian ICP-MS laboratories have been added. Figure 8 has been revised to include measurement and accumulation uncertainties. For the other figures, uncertainty bars are too small to be added (Figs 5, 6) or the resolution of the data shown precludes the addition of error bars (Figs 4, 7). Finally, a paragraph has been added to section 2.2.2 describing interlaboratory reproducibility measurements, for which two supplementary figures have also been added.

Line 30, page 5 and line 3, page 6. Mass resolution (m/dm) is dimensionless. Please, delete "amu".

Done

Lines 4-5, page 6. Please, summarize the method performances in terms of accuracy and difference between blanks and samples values.

As per the previous comment, this information has been added. To summarise here:

A 10 times repetition of the same samples obtain a standard deviation in average of 5% (from 2% to 8% maximum). The blank value is in the order of 40 cps while a samples is in the range of 150 cps using the medium resolution mode. This is an average value since the sensibility of the instrument can change (quite common for the ICP-SFMS) modifying the blank value and the response of the instrument to the sample concentration. During the analysis and external

calibration were run every 20 samples to correct the instrument oscillation. Detection limits, calculated as three times the standard deviation of the blank, were 5 and 50 pgg for I and Br, respectively. Reproducibility was evaluated by repeating measurements of selected samples characterized by different concentration values (between 20 pg g and 400 pg g for I and between 400 and 600 pg g for Br). The residual standard deviation (RSD) was low for both halogens and ranged between 1–2 % and 2–10 % for Br and I, respectively. Detail information can be found in Spolaor et al 2013 (The Cryosphere, 7, 1645–1658 , 2013)

Line 9, page 6. The term "core" has to be referred to DSS0505 and not to DSS1516. Where the DSS1516 snow pit were analyzed? At lines 18-19, page 5, Authors report that DSS1516 snow pit samples were analyzed for Br in Australia. Maybe, in Italy the same samples (or samples from a parallel column in the snow pit) were analyzed even for I and Na, other than Br. In this case, were the Br values compared?

The description of DSS1516 as a core was a mistake and has been removed.

We note in section 2.1.4 "All samples collected during the Casey-Law Dome traverse and from the DSS1516 snowpit were sent to the TRACE laboratory at Curtin University of Technology for bromine analysis." and in section 2.2.1 "Law Dome traverse and DSS1516 snowpit samples were analysed discretely using a Seafast-II autosampler with syringe pump connected to the abovementioned analytical system."

As we have noted above, a laboratory intercomparison has been performed using Greenland snow pit samples and the results are discussed in section 2.2.2 and shown in figures S1 and S2.

Section 2.3. Which samples were analyzed for IC? Were the Na values reported in the manuscript analyzed by IC (soluble fraction) or by ICP-MS (probably total content)? The analyzed Na fraction could play a not-negligible role in evaluating the Br-enr and I-enr fractions, if Na, and not ssNa, is used as sea spray marker. Also for IC measurements, the methods performances (at least for Na and MSA) should be here summarized.

The reviewer is correct that IC typically determines soluble sodium whereas ICP-MS typically determines total Na. The partitioning of sea-salt and mineral dust inputs to Law Dome have been studied in detail by Vallelonga et al. 2004 (Ann. Glac.) as well as Curran et al. (Ann. Glac., 1998). Vallelonga et al., found that sea salts accounted for 98% of impurities in Law Dome snow by mass (average 205 ng/g) whereas mineral dust was just 2% (2.8 ng/g). Therefore, there is little difference between total Na and ssNa for Holocene samples from Law Dome.

Sodium has been determined by ICP-MS in all samples. Additionally, Na has been determined in DSS0506 samples by IC. In fact, comparison of Na concentrations determined by IC and ICP-MS in the DSS0506 samples was the method used to evaluate contamination of the samples (due to cracking of the sample vials) and discard contaminated results. This information has been added to the text.

The IC performance statistics have been added.

Lines 2-3, page 7. Please, reword the sentence.

Done

Lines 7-8, page 8. I think that median is more suitable than geometric mean in evaluating the asymmetry of the data sets, by comparison with the mean values.
Medians have been added

| Sample designation | Time interval | # years | Sodium (ppb) | | | Bromine (ppb) | | | Ln(Br) | | Ln (Br$_{enr}$) | |
|---|---|---|---|---|---|---|---|---|---|---|---|---|
| | | | mean | median | variance | mean | median | variance | mean | variance | mean | variance |
| DSS1516 snowpit | 2015-2016 | 0.8 | 241 | 179 | 91078 | 1.4 | 0.8 | 3.5 | -0.34 | 1.5 | 0.1 | 4.6 |
| DSS1213 core | 1987-2012 | 25 | 115 | 106 | 4355 | 2 | 1.9 | 0.2 | 0.67 | 0.06 | 1.8 | 0.1 |
| DSS0506 core | 1927-1986 | 59 | 77 | 74 | 571 | 3.3 | 2.3 | 9.7 | 0.88 | 0.57 | 2.5 | 0.6 |

Table 1 revised

Line 11, page 8 and following. The calculation of nssBr, nssI, Br-enr and I-enr have to be made by using ssNa, and not total Na, as sea spray marker. I'm aware that, in a coastal site, the majority of the Na content in the snow is originated by sea spray, but also local or long-range dust could give not negligible contributions, at least in particular transport events. As well known, the nssNa fraction (and then ssNa by difference) can be easily evaluated by using Al (if Na is measured by ICP-MS) or nssCa (if Na is measured by IC) as crustal markers and knowing the Al/Na or Ca/Na ratios in the uppermost Earth crust.
As discussed in a previous response, the influence of mineral dust on sodium concentrations at Law Dome is negligible.
Line 12, page 8. Please, change "sodium" in "ssNa".
For the reasons mentioned above, we prefer not to change sodium to 'ssNa' in this case.
Line 15, page 8. Authors are requested adding a short description of meaning and seasonal occurrence of the "bromine explosion" events.
This information has been provided in the text added to the introduction.
Lines 17-18, page 8. The calculation of the enrichment factors of different snow and aerosol components with respect to seawater composition is well established and cannot be attributed to some of the Authors.
We do not imply that this method should be attributed to the authors, rather to indicate previous applications of this calculation with respect to bromine. The text has been amended.
Line 18, page 8. Please, change "Na" in "ssNa".
For the reasons mentioned above, we prefer not to change 'Na' to 'ssNa' in this case.
Line 22, page 8 and following. In my opinion, the tentative explanation of the different variability of Na and Br in the two records (DSS0506 and DSS1213 firn cores) appears to be not convincing. Table 1 and Figure 4 show that DSS1213 Na profile has higher mean values and much higher variability with respect to the DSS0506 record. On the contrary, the DSS1213 Br profile shows a very sharp smoothing of the 3-yr running mean and lower mean values, with respect to the DSS0506 record. A so large Na and Br opposite variability cannot be attributed, in my opinion, to selective (Br, with respect Na) "memory effects" or to a different depth resolution of the analytical methods (melter vs discrete samples). Memory effects are usually related to the matrix and not to single components; besides they could play a smoothing effect (but not as large as for the Br) and cannot increase the variability (as shown by the Na profile). Differences in measurements resolution (continuous melting vs discrete samples) are fully able

to change the data variability, but not in opposite sign for the two components; besides, the different resolution (if not too much large with respect to accumulation rate) should be not able to change the 3-yr mean profiles. Authors are requested to report the estimated depth resolution for the continuous melter system. Finally, snow pit data show very higher Na values and similar Br concentrations (it is difficult to evaluate little differences in a logarithmic scale; Authors are requested to add the snow pit mean values in Table 1), with respect to DSS1213 firn core. How the Authors can explain these patterns? Could the different profiles be caused by different analytical methods in the different laboratories?

Reviewer 1 also expressed similar concerns and the text has been altered to better explain the sources of apparent discrepancy between DSS0506 and DSS1213 records. Following the previous comment of the reviewer, we have added medians and DSS1516 snowpit data to table 1.

As mentioned in our response to reviewer1, the text was not clear in explaining the smoothing of the Br signal is not attributed to the melter, but instead to the ICP-MS sample introduction system used during the continuous melting analysis. Bromine is commonly known to be a "sticky" element for ICP-MS measurements and therefore the instrument requires a specialised cleaning method (with $NH_4OH$, as described in the text). The DSS0506 samples were sampled and measured discretely, hence the sample introduction system could be thoroughly cleaned between each analysis. The DSS1213 core was analysed continuously in a long melting sequence, hence there was limited opportunity to thoroughly clean the sample introduction system. This is the reason why a comparable smoothing is absent in the sodium record.

The depth resolution of the melter system (less than 1 mm) is standard for contemporary CFA systems, and has been added to the text.

Regarding the accuracy of the two instruments used for measurements, these have also been described in our response to reviewer 1 and are treated in section 2.2.2 and supplementary figures S1 and S2.

[Figure]

Supplementary Figure S2

Line 6, page 9. I cannot understand how the variability in the Br-enr data "may act to artificially increase the correlation". Usually, higher variability could cause a loss of correlation or make more difficult the evaluation of a possible correlation between two parameters. Authors are requested to explain their thought.

The sentence was not clear. We are referring to autocorrelation induced by a non-gaussian distribution of the data. The sentence has been rephrased as "With the intention of reducing data autocorrelation, we transform the $Br_{enr}$ data to a gaussian-like distribution using the natural logarithm of $Br_{enr}$ for correlation to FYSI data." and we have plotted data histograms in supplementary figure S3.

Line 17, page 9. The Br-enr – FYSI correlation is poor (max R2 value: 0.32) for the 90-110 °E sector and null for the 110-130 °E sector. This sector selectivity seems to be too large (see, also, I-enr, that shows a completely opposite pattern) and could imply that the correlation between the two parameters (Br-enr or I-enr with FYSI) is weak and possibly covered by other factors, such as atmospheric circulation modes (with an opposite effect for Br-enr and I-enr?). Indeed, a 0.32 R2 value, even if significant at the 99% level, means that just 1/3 of the Br-enr variance can be attributed to changes in FYSI. In my opinion, the R2 values are not sufficient to support the Authors hypothesis (see also my below comments to the Figures 5 and 6).

Please note our opening comment that $Br_{enr}$ and $I_{enr}$ correlations to FYSI have changed as a result of a calculation error in the sea ice algorithm used for the initial submission.

We agree that the $Br_{enr}$-FYSI correlation is not strong but it is comparable to correlation statistics used for sea ice reconstructions at Law Dome based on MSA ($r^2$=0.36, p<0.002) using sea ice extent between 80ºE and 140ºE. Note also that we base our regression calculation on FYSI area rather than maximum or minimum sea ice extent (SIE), which we hypothesize to be most directly representative of the bromine enrichment due to bromine photochemical recycling.

Regarding iodine we note that there are clear indications of summertime re-emission (at both Law Dome and Neumayer station) and hence we are more tentative with our conclusions. We have written:

"Correlation significance levels for $I_{enr}$ are consistently below the 95% level and correlation coefficients are generally low. The strongest correlation of $I_{enr}$ is between the summer-summer (calendar year) averaged signal and the 110-130° E FYSI sector, but this is still below the 95% significance level."

In response to similar concerns of reviewer1, we have "scaled back" the conclusions regarding the use of bromine for reconstruction of sea ice at Law Dome.

Line 20, page 9. The MSA-FYSI correlation in the 80-140 °E sector cannot support the Br-enr – FYSI correlation because the last correlation is highly sector selective and the 80-140 °E sector covers a sector (110-130 °E) in which the Br-enr – FYSI correlation is completely absent.

We have amended the text accordingly

Lines 24-30, page 9. As before discussed (my comments to line 17, page 9), Authors attribute to several possible "noise effects" the Br-enr – FYSI correlation variability as a function of summer-summer or winter-winter intervals. In my opinion, the correlation is always poor and R2 values depend on too much factors to be confident. The last sentence (lines 29-30) is not supported by the data.

The last sentence has been removed.

Lines 14-15, page 10. These limitations in the comparison of MSA and Br-enr temporal trends are correct, but we have to consider that we are comparing 11-yr mean profiles. Therefore, some limiting factors, especially the different seasonal pattern, surely play a minor or null role.

The text has been changed accordingly:.

Due to the 11-year smoothing applied to the data, influences of seasonal patterns, factors influencing biological growth, relations to sea ice and transport efficacy should be minimised for the comparison of bromine enrichment and MSA trends at Law Dome

Line 21 and line 23, page 10. I cannot see in Figure 5 a significant increase, with respect to the noisy baseline along the multi-decadal trend, of MSA during the periods 1920-30 and 1975-85. For instance, the MSA profile in the period 1955-67 could show similar positive anomalies. The only significant increase is related to the period 1940- 55.

We agree and have changed the text accordingly.

Line 22, page 10. I agree that MSA and Br-enr profiles show a common multi-decadal trend (a slight decrease), but the single common feature, around 1940-1955, shows peaks shifted of about 4 years. Besides, the trend in the period 1955-80 is opposite. Even neglecting the large 1970-90 Br-enr peak, which is not evident in MSA profile, the 1955-70 trends of Br-enr (increasing) is opposite to that of MSA (decreasing). I think that the agreement between the two parameters is weak and Authors should try to explain the observed differences in the temporal shift of the 1940-50 peaks and in the 1955-70 opposite trends.

We agree with the reviewer (and also with respect to the comments of reviewer1) that there is not a constant agreement between the MSA and Br records shown in Figure 5. As we have noted in the text, both data indicate a peak in the 1940's and a small decreasing trend over the 20th century, but diverge after 1955. Given that the data are smoothed by 11-year running means, we are not necessarily concerned by a 4-year difference in the peaks of the two data sets in the 1940's [1949 MSA, 1945 ln(Br$_{enr}$)].

As the reviewer has noted, only a third of the variance of Br can be attributed to FYSI, so it is clear that there may be influences that are not related to sea ice variability. With regard to the divergence between 1955 and 1970, we have written:

"Bromine and MSA both point toward greater sea ice area during the period from 1945 to 1950 but diverge between 1955 and 1970. The cause for this divergence is not yet known, but before speculating on a possible cause, these trends should be confirmed by measurements of other snow and ice samples from Law Dome as well as other sectors of the East Antarctic coast. The possible influence of multidecadal-scale climate variability, such as the Interdecadal Pacific Oscillation (IPO), on the bromine record is discussed in detail in section 3.4, but will briefly

be considered here. IPO forcing of Antarctic sea ice area has been demonstrated at decadal timescales (Meehl et al., 2016), with the negative IPO phase triggering SLP and near surface wind changes that can influence sea ice expansion, storm tracks and potentially nutrient supply to DMS-producing algal communities."

Lines 25-26, page 10. I have some perplexities also concerning the comparison between FYSI and Br-enr. Unfortunately, the period covered by satellite measurements is short and a reliable comparison is difficult. However, I can see two clear evidences that should be explained. By observing the Br-enr large peak around 1970-90, we have to note that while it is correct that the highest value is synchronous with the 1982 large positive anomaly in the FYSI, its temporal evolution does not follow the FYSI dynamics. Indeed, the Br-enr peak show an abrupt increase when FYSI positive anomalies are not marked (unfortunately, FYSI 1977 and 1978 satellite data are missing, but 1976 and 1979-80 data show null or slightly negative anomalies). Besides, after the 1981 peak, Br-enr quickly decreases, while FYSI shows relevant positive anomalies until 1985. Finally, almost continuous FYSI negative anomalies from 1993 to 2010 do not cause negative peaks in the Br-enr profile that, on the contrary, shows a continuous and clear increase. I think the Authors should reconsider their assessments and better discuss (and, possibly, interpret) the complex FYSI-Br-enr relationship. As the Authors report in the next section, the two Br-enr peaks (around 1945 and 1981) could be related to changes in atmospheric circulation modes (e.g., changes in the IPO) that could include (but not only) changes in sea ice dynamics.

We respect the reviewers clear interest in the applicability of Br to reconstructing FYSI, but we are hesitatant to expect a perfect year-by-year agreement between $Br_{enr}$ and FYSI area. We have observed a similar correlation coefficient for Br and FYSI as Curran et al. found for MSA and SIE (Sea Ice Extent), which may suggest that other factors, such as meteorological noise, deposition variability and ice core representablity may account for the other 70% of signal variability.
In line with this and other comments from both reviewers, we have made a more conservative evaluation of $Br_{enr}$ as a proxy of Law Dome sea ice variability.

Lines 6-7, page 11. The correlation coefficient between I-enr and FYSI for summer- summer 110-130 ◦E sector (R2 = 0.42) is higher than that of Br-enr for 90-110 ◦E (R2 = 0.32), even if the p-value is slightly lower (<0.01, with respect to <0.001, but anyway significant). Authors should discuss why the correlation between I-enr and FYSI is so sector selective, and opposite of that, similarly sector selective, between Br-enr and FYSI.

Please note our opening comment that $Br_{enr}$ and $I_{enr}$ correlations to FYSI have changed as a result of a calculation error in the sea ice algorithm used for the initial submission.
As we note in the text (and following suggestions by reviewer1) halogen chemistry is complex and there are many chemical processes that are still to be understood, especially with regard to iodine. Considering that it has been observed that iodine is reemitted from the snow surface, and is retained in snowpack only in the winter, we also find it surprising that there is any

correlation between FYSI and $I_{enr}$ at all. Using our corrected sea ice area calculations, we find that some years (1973, 1975) of the FYSI record should have been excluded, as full year-round observations were not available. Consequently, we do not find a significant correlation with $I_{enr}$ for any sea ice sector or $I_{enr}$ seasonal pattern. As we have written in the text, halogen chemistry is a topic of ongoing research and an accurate interpretation of the data is best served by building up an array of records from along the East Antarctic coast.

Line 10, page 11. A correlation with R2 = 0.27, even if the value is statistically significant, is really too poor and cannot demonstrate that the two parameters are correlated (just 1 of the variance of a parameter is explained by the variability of the other). However, by observing figure 6, the two profiles are very similar. Maybe, the correlation is poor because there are temporal shifts between the peaks of the two records. Indeed, the 1945 Br-enr peak leads the I-enr peak and the opposite pattern is visible for the 1980 Br-enr peak. In my opinion, the Br-enr – I-enr relationship deserves an improved discussion.

We agree with the reviewer that the correlation between $I_{enr}$ and $Br_{enr}$ is statistically weak but visually distracting. We have expanded our consideration of possible co-incident causes leading to such similar variability between the two measures, while being cautious that the correlation is poor.

Lines 13-18, page 11. The relationship of I-enr and Br-enr with IPO is potentially very interesting. Unfortunately, Authors barely touches on the topic. The Authors should improve the discussion and evaluate how the IPO changes can affect the Br and I emissions or transport processes. For instance, why positive-to-negative and negative- to-positive IPO phase changes cause the same effects on I-enr and Br-enr profiles? Which are the relationships between IPO phases and atmospheric circulation around Antarctica or sea-ice dynamics?

We have expanded our description of links between IPO and Antarctic atmosphere-ocean-sea ice variability and relevance to coastal East Antarctica:

"The IPO is a low frequency climate mode related to the El Niño-Southern Oscillation which operates on multidecadal timescales. It affects climate variability at the multidecadal scale across and beyond the Pacific Basin (Power et al., 1999; Vance et al., 2015). Impurities deposited at Law Dome have been demonstrated to faithfully reflect IPO variability (Vance et al., 2015) and reanalysis data indicates a strong IPO signal in the Indian Ocean (Vance et al., 2016). Furthermore, recent work has demonstrated an IPO forcing of Antarctic sea ice area at decadal timescales, with the late 1990's shift to a negative IPO phase triggering SLP and near surface wind changes that are conducive and consistent with an expansion in sea ice in all seasons across multiple regions of the Antarctic seasonal ice zone (Meehl et al., 2016). Thus the overall correlation between iodine and bromine enrichment may be linked to decadal-scale states of the atmosphere-ocean-sea ice system in the Indian sector of the Southern Ocean. It must be noted that, in addition to larger-scale influences of atmospheric transport and ocean-related sea ice variability, both Ienr and Brenr are calculated using Na as an indicator of sea salt content in the samples. The

possibility of an IPO-related signal, transmitted through Na concentrations, cannot be discounted from contributing to the apparent correlation of Ienr and Brenr in DSS0506 core samples."

Section 3.5. Maybe this section should be moved just after (or inside) section 3.1. No novelty on the seasonal pattern of Br is here reported, with respect to previous results on shorter data series. The tricky dephasing between spring Br explosion and Br summer maximum in the snow is not explained (and I agree that this pattern has to be in deep studied).

We prefer to keep the discussion of seasonality as a stand-alone section, mostly because the "previous results on shorter data series" that the reviewer refers to is just a four-year period from 1910 to 1914. Furthermore, those data were measured from discrete samples of the DSS0506 core and in this way we are able to present an independent seasonality study, from a different core sampled and measured independently, with better temporal control and better statistics.
Regarding the lag between spring bromine explosion and summer $Br_{enr}$ peak, we write the following:
"Satellite observations of atmospheric BrO in polar regions suggest an early spring peak in bromine activity in Antarctica (Spolaor et al., 2014), thereby implying that additional processes may be occurring in the snowpack during the summer after the peak atmospheric bromine explosion has occurred. While snowpack remobilisation at Law Dome is minimal, it might be the case that photochemically-driven heterogenous recycling of bromine occurs in the snowpack after the springtime occurrence of the bromine explosion. This effect requires further investigation, from satellite and ground-based observations to weekly surface snow sampling, in order to be fully characterised and understood."

Line 16, page 12. Figure 8 show fluxes and not concentrations. Na fluxes are very higher in the final 25 km, with respect to more coastal sites.
The text has been changed accordingly.

Line 20, page 12. This Na and Br pattern is interesting and should be enlightened. Higher fluxes in higher snow-accumulation sites mean that Na and Br deposition occurs mainly by wet-deposition, while dry deposition could be negligible. This fact can have implications in ice-core studies.

We appreciate that the reviewer finds this novel spatial-transect study of interest. As we note in the text, the samples were collected during a traverse from Casey station to Law Dome summit, and hence have been collected on the down-wind (or lee-side) of Law Dome. For a thorough evaluation of deposition characteristics of Na and Br at Law Dome it is important to sample further east, on the up-wind side of Law Dome.

Conclusions. This section should be revised accordingly to the suggested manuscript changes.
Done

---

## Author Comment (AC3) · 7 Dec 2016

1) Please note an error in the text: The altitude of Law Dome Summit is 1370 m, not 1310 m as written in the revised text (page 4, line 20).

2) The data are now archived on PANGAEA: https://doi.pangaea.de/10.1594/PANGAEA.868431 The link to NOAA Paleoclimate database will be added when the data is archived.